# Cysteine dioxygenase 1 is a metabolic liability for non-small cell lung cancer

**Yun Pyo Kang[1], Laura Torrente[1], Aimee Falzone[1], Cody M Elkins[1], Min Liu[2], John M Asara[3,4], Christian C Dibble[5,6], Gina M DeNicola[1]***

[1]Department of Cancer Physiology, H Lee Moffitt Cancer Center and Research Institute, Tampa, United States; [2]Proteomics and Metabolomics Core Facility, H Lee Moffitt Cancer Center and Research Institute, Tampa, United States; [3]Division of Signal Transduction, Beth Israel Deaconess Medical Center, Boston, United States; [4]Department of Medicine, Harvard Medical School, Boston, United States; [5]Department of Pathology and Cancer Center, Beth Israel Deaconess Medical Center, Boston, United States; [6]Department of Pathology, Harvard Medical School, Boston, United States

**Abstract** NRF2 is emerging as a major regulator of cellular metabolism. However, most studies have been performed in cancer cells, where co-occurring mutations and tumor selective pressures complicate the influence of NRF2 on metabolism. Here we use genetically engineered, non-transformed primary murine cells to isolate the most immediate effects of NRF2 on cellular metabolism. We find that NRF2 promotes the accumulation of intracellular cysteine and engages the cysteine homeostatic control mechanism mediated by cysteine dioxygenase 1 (CDO1), which catalyzes the irreversible metabolism of cysteine to cysteine sulfinic acid (CSA). Notably, *CDO1* is preferentially silenced by promoter methylation in human non-small cell lung cancers (NSCLC) harboring mutations in KEAP1, the negative regulator of NRF2. CDO1 silencing promotes proliferation of NSCLC by limiting the futile metabolism of cysteine to the wasteful and toxic byproducts CSA and sulfite ($SO_3^{2-}$), and depletion of cellular NADPH. Thus, CDO1 is a metabolic liability for NSCLC cells with high intracellular cysteine, particularly NRF2/KEAP1 mutant cells.
DOI: https://doi.org/10.7554/eLife.45572.001

*For correspondence:
Gina.DeNicola@moffitt.org

**Competing interests:** The authors declare that no competing interests exist.

## Introduction

NRF2 (Nuclear factor-erythroid 2 p45-related factor two or NFE2L2) is a stress-responsive cap'n'collar (CNC) basic region leucine zipper (bZIP) transcription factor that directs various transcriptional programs in response to oxidative stress. Under basal conditions, NRF2 is kept inactive through binding to its negative regulator KEAP1 (Kelch-like ECH-associated protein), which is a redox-regulated substrate adaptor for the Cullin (Cul)3-RING-box protein (Rbx)1 ubiquitin ligase complex that directs NRF2 for degradation (*Kobayashi et al., 2004*). KEAP1 is the major repressor of NRF2 in most cell types, which is supported by the evidence that disruption of *Keap1* in the mouse increased the abundance and activity of Nrf2 (*Wakabayashi et al., 2003*). NRF2 plays a critical role in tumor initiation and progression in response to oncogenic signaling and stress (*DeNicola et al., 2011*; *Todoric et al., 2017*). Further, NRF2 and KEAP1 mutations are common in many cancers and lead to impaired NRF2 degradation and constitutive NRF2 accumulation (*Ohta et al., 2008*; *Shibata et al., 2008*), thereby promoting glutathione (GSH) synthesis, detoxification of reactive oxygen species (ROS) and proliferation.

While the role of NRF2 in ROS detoxification is well established, novel roles of NRF2 in the regulation of cellular metabolism have been recently identified. NRF2 promotes the activity of the

**eLife digest** Cancers form in humans and other animals when cells of the body develop mutations that allow them to grow and divide uncontrollably. The set of chemical reactions happening inside cancer cells, referred to as "metabolism", can be very different to metabolism in the healthy cells they originate from. Some of these differences are directly caused by mutations, while others are a result of the environment surrounding the cancer cells as they develop into a tumor.

A protein called NRF2 is often overactive in human tumors due to mutations in its inhibitor protein KEAP1. Previous studies have shown that NRF2 changes the metabolism of cancer cells by switching specific genes on or off. However, since cancer cells also have other mutations that could mask or amplify some of the effects of NRF2, the precise role of this protein in metabolism remains unclear.

To address this question, Kang et al. generated mice that could switch between producing the normal KEAP1 protein or a mutant version that is unable to inhibit NRF2. The mouse model was then used to examine the immediate effects of activating the NRF2 protein. This revealed that NRF2 altered how mouse cells used a molecule called cysteine, which is required to make proteins and other cell components. When NRF2 was active, some of the cysteine molecules were converted into two wasteful and toxic particles by an enzyme called CDO1.

Kang et al. found that inactivating CDO1 in human lung cancer cells prevented these wasteful particles from being produced. This allows cancer cells to grow more rapidly, and may explain why human tumors generally evolve to shut down CDO1.

The findings of Kang et al. show that not all of the changes in metabolism caused by individual mutations in cancer cells help tumors to grow. As a tumor develops it may need to acquire further mutations to override the negative effects of these changes in metabolism. In the future these findings may help researchers develop new therapies that reactivate or mimic CDO1 to limit the growth of tumors.

DOI: https://doi.org/10.7554/eLife.45572.002

pentose phosphate pathway to support the production of NADPH and nucleotides (*Mitsuishi et al., 2012*; *Singh et al., 2013*). Further, NRF2 promotes serine biosynthesis to support GSH and nucleotide production (*DeNicola et al., 2015*). These metabolic programs support cell proliferation and tumor growth but not all metabolic consequences of NRF2 activation are favorable. Although uptake of cystine $(CYS)_2$ via the xCT antiporter (system $x_c^-$) promotes GSH synthesis and antioxidant defense (*Sasaki et al., 2002*), it also induces glutamate export and limits glutamate for cellular processes (*Sayin et al., 2017*). NRF2 suppresses fatty acid synthesis to conserve NADPH, which may antagonize proliferation (*Wu et al., 2011*). Importantly, the activity of these metabolic pathways may be influenced by co-occurring mutations found in the model systems used for study, such as LKB1 mutations, which commonly co-occur with KEAP1 mutations and influence NADPH levels (*Jeon et al., 2012*; *Skoulidis et al., 2015*). Further, NRF2 directs distinct transcriptional programs under basal and stress-inducible conditions (*Malhotra et al., 2010*), complicating the interpretation of its effects on cellular metabolism.

To examine the immediate consequence of constitutive NRF2 stabilization on cellular metabolism in non-transformed cells, we generated a genetically engineered mouse model expressing the $KEAP1^{R554Q}$ loss-of-function mutation found in human lung cancer. Using this model, we have examined the control of cellular metabolism by NRF2 in mouse embryonic fibroblasts (MEFs) and find that NRF2 promotes the accumulation of intracellular cysteine (CYS) and sulfur-containing metabolites, including GSH and the intermediates of the taurine (TAU) biosynthesis pathway cysteine sulfinic acid (CSA) and hypotaurine (HTAU). Entry of CYS into the TAU synthesis pathway was mediated by cysteine dioxygenase 1 (CDO1), which was elevated in $KEAP1^{R554Q}$ MEFs. TAU synthesis is initiated by the irreversible metabolism of CYS by CDO1 to CSA, which is then decarboxylated by cysteine sulfinic acid decarboxylase (CSAD) to HTAU. In turn, HTAU is non-enzymatically converted to TAU, or CSA is transaminated by the cytosolic aspartate aminotransferase (GOT1) to produce β-sulfinyl pyruvate, which spontaneously decomposes to pyruvate and sulfite $(SO_3^{2-})$. At the organismal level,

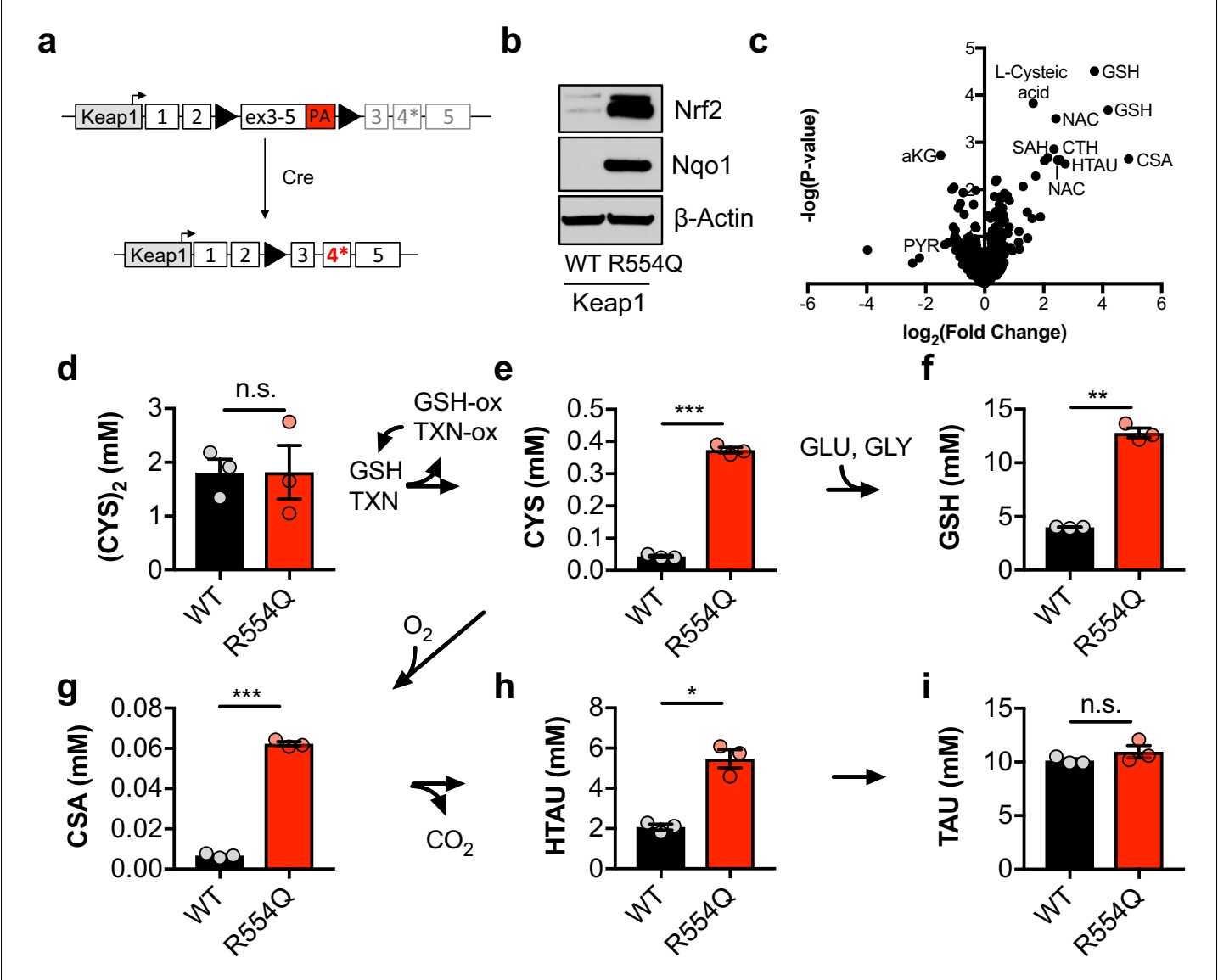

**Figure 1.** Nrf2 promotes the accumulation of intracellular cysteine and sulfur-containing metabolites. (a) Schematic of the murine Keap1[R554Q] allele. The Keap1[R554Q] allele was created by inserting a loxP-flanked wild-type Keap1 cDNA containing exons 3–5 upstream of a R554Q mutation in the endogenous exon 4 of the Keap1 gene. Prior to recombination, wild-type Keap1 protein is expressed due to splicing into the cDNA containing exons 3–5, and is referred to as Keap1[WT]. Following Cre-mediated excision of the loxP-flanked cargo, Keap1[R554Q] is expressed at physiological levels and is referred to as Keap1 [R554Q]. PA, poly A signal. (b) Western blot analysis of Nrf2, Nqo1, and β-Actin levels following Nrf2 stabilization in wild-type (WT) vs. homozygous Keap1[R554Q/R554Q] (R554Q) MEFs. (c) LC-HRMS metabolomics profiling of Keap1[R554Q/R554Q] MEFs compared to Keap1[WT/WT] MEFs. GSH, glutathione. CSA, cysteine sulfinic acid. HTAU, hypotaurine. CTH, cystathionine. NAC, N-acetyl cysteine. SAH, S-adenosyl homocysteine. aKG, α-ketoglutarate. PYR, pyruvate. N = 3, representative of 2 individual MEF lines. (d–i) Quantitation of cystine ($[CYS]_2$, d), cysteine (CYS, e), glutathione (GSH, f), cysteine sulfinic acid (CSA, g), hypotaurine (HTAU, h) and taurine (TAU, i) levels in Keap1[R554Q/R554Q] MEFs compared to Keap1[WT/WT] MEFs. Cysteine and glutathione were derivatized with N-ethylmaleamide (NEM) to prevent oxidation during extraction here and for all quantification experiments. TXN, thioredoxin. TXN-ox, oxidized thioredoxin. GSH-ox, oxidized glutathione. GLU, glutamate. GLY, glycine. N = 3.
DOI: https://doi.org/10.7554/eLife.45572.003

The following source data and figure supplements are available for figure 1:

**Source data 1.** Nrf2 promotes the accumulation of intracellular cysteine and sulfur-containing metabolites.
DOI: https://doi.org/10.7554/eLife.45572.005

**Figure supplement 1.** Cre infection does not induce taurine pathway activity in MEFs.
DOI: https://doi.org/10.7554/eLife.45572.004

**Figure supplement 1—source data 1.** Cre infection does not induce taurine pathway activity in MEFs.

*Figure 1 continued on next page*

Figure 1 continued

DOI: https://doi.org/10.7554/eLife.45572.006

decarboxylation of CSA via CSAD predominates over transamination by GOT1 (*Weinstein et al., 1988*). By contrast, lung cancer cell lines accumulated significant CYS due to epigenetic silencing of the *CDO1* locus. CDO1 re-expression antagonized proliferation and promoted the metabolism of CYS to CSA, but surprisingly most CSA was exported from cells or transaminated to produce toxic $SO_3^{2-}$. Further, continual $(CYS)_2$ reduction to replenish the CYS pool impaired NADPH-dependent cellular processes. These results demonstrate that CDO1 antagonizes the proliferation of lung cancer cells with high intracellular CYS and its expression is selected against during tumor evolution.

## Results

### NRF2 promotes the accumulation of sulfur-containing metabolites

To evaluate how constitutive NRF2 activity reprograms metabolism, we generated a genetically engineered, conditional knock-in mouse model of the cancer mutation KEAP1$^{R554Q}$ (*Figure 1A*). Mutations at this residue prevent the association of KEAP1 with NRF2, thereby stabilizing NRF2 and inducing the expression of NRF2 target genes (*Hast et al., 2014*). We inserted a loxP-flanked wild-type *Keap1* cDNA upstream of the R554Q mutation in exon four in the endogenous *Keap1* gene. Prior to exposure to Cre recombinase, wild-type Keap1 protein is expressed. Following Cre-mediated excision of the loxP-flanked cargo, mutant Keap1$^{R554Q}$ is expressed at physiological levels, thus recapitulating the genetic events of human NSCLC and allowing for the interrogation of the consequences of Keap1$^{R554Q}$ expression in an isogenic system. Mouse embryonic fibroblasts (MEFs) harboring this allele were derived to evaluate the consequence of Keap1$^{R554Q}$ expression in primary cells. The expression of homozygous *Keap1$^{R554Q}$* led to Nrf2 accumulation and increased expression of the Nrf2 target Nqo1 (*Figure 1B*). We performed non-targeted metabolomics to identify metabolite alterations in these cells and found that the most abundant metabolites following Nrf2 accumulation are sulfur-containing metabolites derived from CYS (*Figure 1C*), while infection of wild-type MEFs with adenoviral Cre did not significantly alter metabolite levels (*Figure 1—figure supplement 1A*). To interrogate cysteine metabolism in more detail, we performed targeted metabolomics to quantify the concentration of intracellular CYS and its downstream metabolites (*Figure 1—figure supplement 1B*). As expected, Nrf2 promoted an increase in intracellular CYS and its downstream metabolite GSH (*Figure 1D–F*), consistent with previous observations that NRF2 promotes the uptake of $(CYS)_2$ and the synthesis of GSH (*Sasaki et al., 2002*; *Wild et al., 1999*). Surprisingly, we also observed a significant increase in intermediates of the TAU biosynthesis pathway, including CSA and HTAU (*Figure 1G–I*). Importantly, HTAU is a highly abundant metabolite and the increase of HTAU was similar to the increase of GSH in the millimolar range (*Figure 1F,H*), suggesting that entry into the TAU biosynthesis pathway may represent a significant percentage of total CYS usage. Collectively, these results indicate that NRF2 promotes the accumulation of intracellular cysteine and entry of cysteine into multiple downstream pathways.

### Nrf2 promotes the entry of cysteine into the taurine synthesis pathway via Cdo1 in non-transformed, primary MEFs

The significant accumulation of intracellular CYS and TAU synthesis intermediates led us to hypothesize that Nrf2 promotes the accumulation of Cdo1 protein, which is stabilized following CYS accumulation due to a loss its ubiquitination and degradation (*Dominy et al., 2006*). We observed a robust increase in Cdo1 protein in Keap1$^{R554Q}$ MEFs compared to Keap1$^{WT}$ MEFs in the absence of an increase in mRNA expression (*Figure 2A,B*), consistent with the known mechanism of Cdo1 regulation. To examine whether Cdo1 mediates CYS metabolism to CSA and HTAU, and whether this limits the use of CYS for GSH synthesis, we deleted Cdo1 with CRISPR/Cas9, followed by infection with empty or Cre expressing adenovirus to generate Cdo1-deficient, isogenic Keap1$^{WT}$ and Keap1$^{R554Q}$ MEFs. Western analysis of Cdo1 protein revealed a significant reduction of Cdo1 expression in Keap1$^{R554Q}$ MEFs, although the already low Cdo1 levels did not change significantly in Keap1$^{WT}$ MEFs (*Figure 2C*). We performed quantitative $^{13}C_6$-cystine [$(CYS)_2$] tracing to examine the entry of

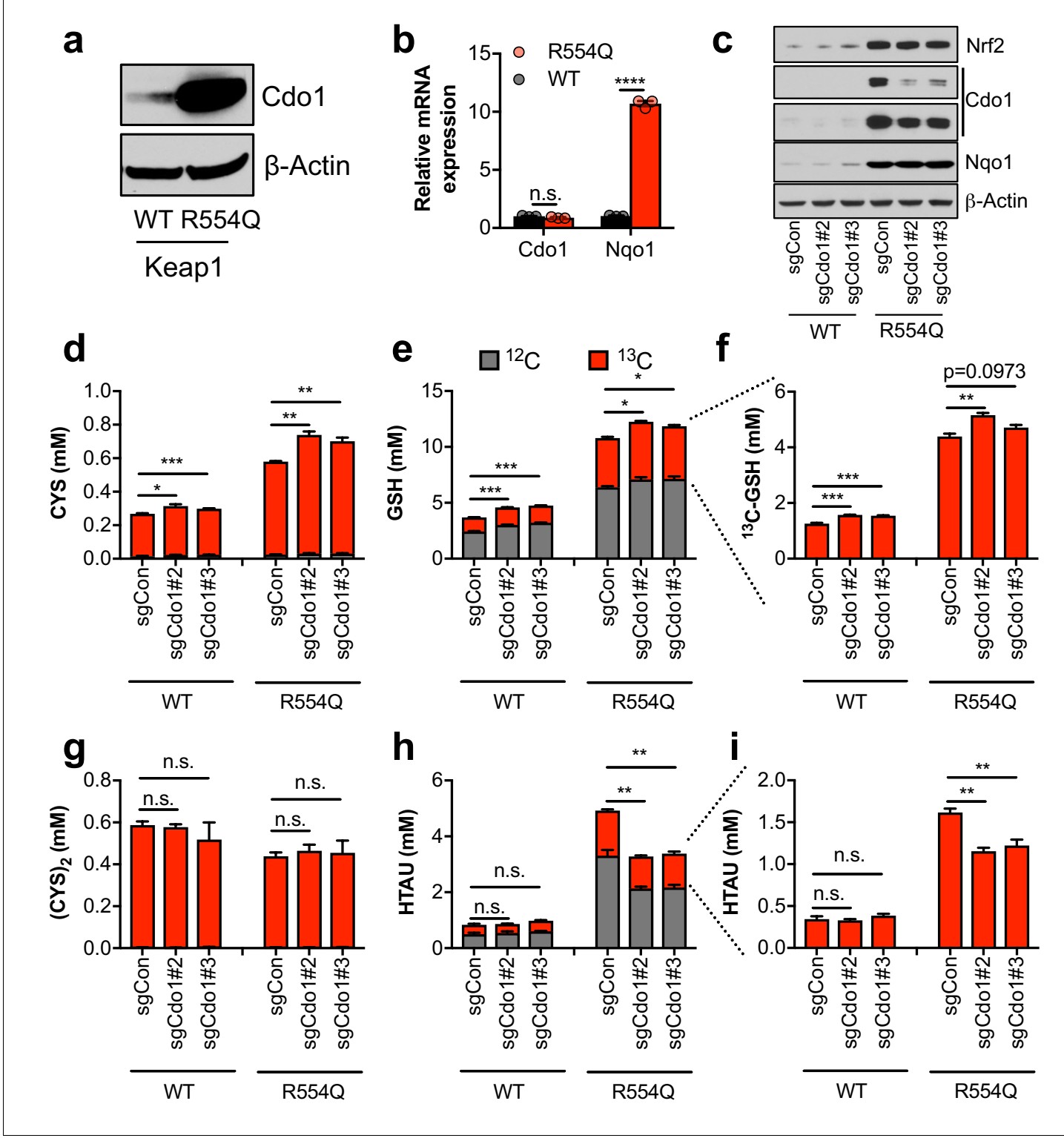

**Figure 2.** Nrf2 promotes the accumulation of cysteine dioxygenase (Cdo1) to promote entry of cysteine into the taurine synthesis pathway. (a) Western blot analysis of Cdo1 and β-Actin levels following Nrf2 stabilization in wild-type (WT) vs. homozygous Keap1[R554Q/R554Q] (R554Q) MEFs. (b) Real-time PCR analysis of Cdo1 and Nqo1 mRNA levels following Nrf2 stabilization in wild-type (WT) vs. homozygous Keap1[R554Q/R554Q] (R554Q) MEFs. mRNA expression was normalized to β-Actin expression, followed by normalization to WT. N = 3. (c) Western blot analysis of Nrf2, Cdo1, Nqo1 and β-Actin levels following expression of sgControl (sgCON) or Cdo1 deletion (sgCdo1 #2 and #3) with CRISPR/Cas9 in primary wild-type (WT) and homozygous

*Figure 2 continued on next page*

*Figure 2 continued*

Keap1$^{R554Q/R554Q}$ (R554Q) MEFs. (d–i) Quantitation of cysteine (CYS, (d), glutathione (GSH, (e,f), cystine ([CYS]$_2$,(g), and hypotaurine (HTAU, (h,i) total levels and $^{13}$C-labeling from $^{13}$C-cystine in cells from (c). $^{13}$C-label is shown in red, while $^{12}$C-label is gray. Cells were labeled for 4 hr. N = 3.

DOI: https://doi.org/10.7554/eLife.45572.007

The following source data is available for figure 2:

**Source data 1.** Nrf2 promotes the accumulation of cysteine dioxygenase (Cdo1) to promote entry of cysteine into the taurine synthesis pathway.

DOI: https://doi.org/10.7554/eLife.45572.008

CYS into GSH and TAU synthesis and found that depletion of Cdo1 inhibited HTAU synthesis from CYS (*Figure 2H,I*). However, HTAU labeling was not completely abolished, which may be explained by incomplete Cdo1 deletion or Cdo1-independent HTAU synthesis from CYS via CoA breakdown, which cannot be distinguished by this method. By contrast, the total CYS and GSH levels as well as GSH labeling from $^{13}$C$_6$ labeled (CYS)$_2$ were modestly increased by Cdo1 depletion, without any change in CYS$_2$ levels (*Figure 2D–G*). These results demonstrate that Cdo1 accumulation in Keap1$^{R554Q}$ MEFs promotes CYS entry in into the TAU synthesis pathway, and modestly limits CYS accumulation and GSH synthesis.

## CDO1 is preferentially silenced in KEAP1 mutant NSCLC and antagonizes proliferation

The limitation of CYS availability by CDO1 suggests that this enzyme may antagonize NRF2-dependent processes in cancer. Thus, we hypothesized that the CDO1-mediated CYS homeostatic control mechanism might be deregulated in NSCLC, allowing enhanced CYS entry into GSH synthesis and other pathways. To evaluate this possibility, we examined the expression of *CDO1* in NSCLC patient samples from The Cancer Genome Atlas (TCGA). CDO1 mRNA expression was significantly lower in lung adenocarcinoma samples compared to normal lung (*Figure 3A*), which was associated with *CDO1* promoter methylation (*Figure 3B*) and poor prognosis (*Figure 3—figure supplement 1A*). Methylation was strongly correlated with mRNA expression across patient samples (*Figure 3—figure supplement 1B*). Interestingly, the incidence of *CDO1* promoter methylation was significantly higher and its mRNA expression significantly lower in KEAP1 mutant lung adenocarcinoma compared to wild-type (*Figure 3A,B*), and NRF2 activity high lung adenocarcinoma compared to NRF2 low (*Figure 3—figure supplement 1C*), suggesting that CDO1 expression confers a selective disadvantage in the context of NRF2 accumulation. CDO1 protein expression was undetectable in a panel of NSCLC cell lines with the exception of H1581 cells (*Figure 3C*), and treatment with the DNMT inhibitor decitabine restored CDO1 mRNA expression (*Figure 3—figure supplement 1D*). These results indicate that CDO1 epigenetically silenced by promoter methylation in NSCLC cell lines and patient samples.

To investigate the NRF2-dependent regulation of CDO1 protein in NSCLC, we generated a doxycycline-inducible lentiviral expression system to reintroduce GFP, CDO1$^{WT}$ or a catalytically inactive CDO1 mutant (Y157F, *Ye et al., 2007*) at single copy into the panel of NRF2$^{LOW}$ and NRF2$^{HIGH}$ NSCLC cell lines (*Figure 3D, Figure 3—figure supplement 2*). The level of CDO1 protein expression in these cells was similar with the physiological Cdo1 levels in mouse lung and liver (*Figure 3—figure supplement 2B*), with liver being one of the highest CDO1-expressing tissues that is responsible for supplying TAU to the body (*Stipanuk et al., 2015*). We find that CDO1 accumulated to higher levels in NRF2$^{HIGH}$ cells than NRF2$^{LOW}$, although accumulation was observed in many NRF2$^{LOW}$ cell lines as well (*Figure 3D*). We investigated the association with intracellular CYS levels across the panel of parental cell lines and found a strong association between the level of CDO1 accumulation and intracellular CYS levels but not with the level of *CDO1* mRNA expressed from our inducible promoter system (*Figure 3D, Figure 3—figure supplement 2A*), which is consistent with our findings in Keap1$^{R554Q}$ MEFs (*Figure 2A,B*). Consistent with MEFs, we also find that deletion of endogenous CDO1 in H1581 cells, the only NSCLC line with detectable CDO1 expression, promoted CYS accumulation, demonstrating that CDO1 functions to limit intracellular CYS in lung cells as well (*Figure 3E*).

To directly examine the effect of NRF2 on CDO1 expression in NSCLC cell lines, we used multiple isogenic cell systems. First, we used NRF2-deficient A549 cells (*Torrente et al., 2017*), in which we

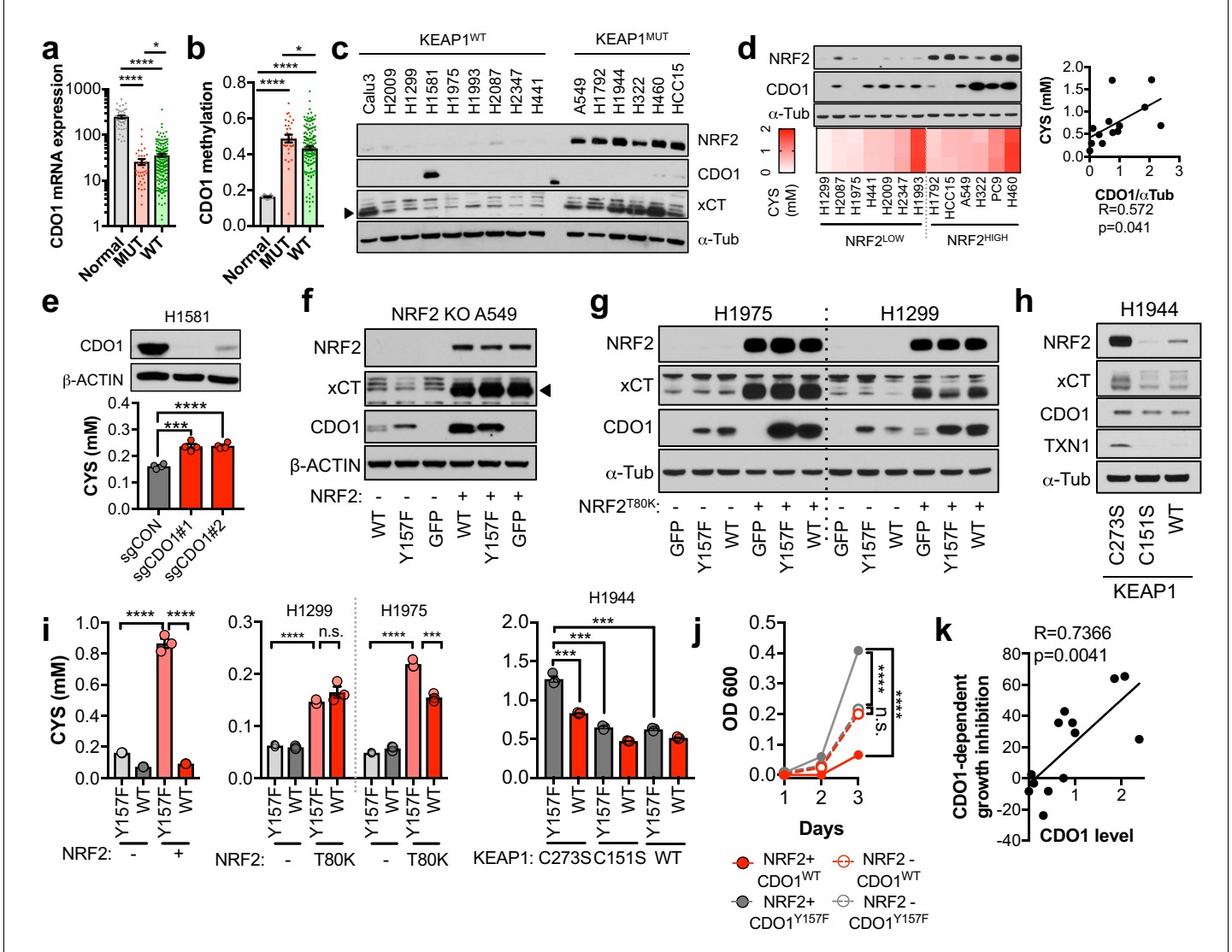

**Figure 3.** CDO1 is preferentially silenced in KEAP1 mutant NSCLC and antagonizes proliferation. (a) CDO1 mRNA expression of normal lung (Normal) or or KEAP1 wild-type (WT) and mutant (MUT) lung adenocarcinoma patient tumor samples. Normal, N = 45. MUT, N = 39, WT, N = 189. (b) Methylation of the *CDO1* promoter in samples from normal lung (Normal) or KEAP1 wild-type (WT) and mutant (MUT) lung adenocarcinoma patient tumor samples. Normal, N = 28. MUT, N = 29, WT, N = 154. (c) Western blot analysis of NRF2, CDO1, xCT and α-Tubulin levels in KEAP1 wild-type and KEAP1 mutant NSCLC cell lines. Arrowhead denotes specific band. (d) (Top) Western blot analysis of NRF2, CDO1 and α-Tubulin expression in NRF2$^{LOW}$ and NRF2$^{HIGH}$ (KEAP1 mutant: H1792, HCC15, A549, H322, and H460) NSCLC cell lines expressing CDO1$^{Y157F}$. (Bottom) Intracellular cysteine concentration of the parental cell lines (minus doxycycline). (Right) Correlation between CDO1 protein levels and intracellular cysteine concentrations. CDO1 protein was normalized to α-Tubulin. N = 13. (e) (Top) Western blot analysis of CDO1 and β-ACTIN expression in H1581 cells following expression of control (sgCON) or CDO1-targeting sgRNAs (sgCDO1 #1 and #2) and Cas9. (Bottom) Intracellular cysteine concentration of the same cells. N = 4 replicates/group. (f–h) Western blot analysis of NRF2, CDO1, xCT, TXN1, α-Tubulin and β-ACTIN levels following re-expression of CDO1$^{WT}$ (WT), CDO1$^{Y157F}$ (Y157F), or GFP in (f) NRF2 knockout A549 cells reconstituted with either pLX317 empty (-) or pLX317-NRF2 (+), (g) H1975 and H1299 cells expressing either pLX317 empty (-) or pLX317-NRF2$^{T80K}$ (+), or (h) H1944 cells reconstituted with inactive pLenti-KEAP1 (C273S), super repressor KEAP1 (C151S) or wild-type KEAP1 (WT). (i) Analysis of cysteine levels in the cells from (e–g). N = 3 replicates/group. (j) Analysis of the proliferation of cells from (f). Cells were collected on the indicated days, stained with crystal violet and their absorbance at 600 nm determined. N = 3 replicates/group. (k) Analysis of proliferation of NSCLC cells expressing CDO1$^{WT}$ and correlation with CDO1 protein expression. Cells were collected after 3 days, and CDO1 dependent growth inhibition was determined by taking the ratio of CDO1$^{WT}$ / CDO1$^{Y157F}$ cell quantity. For individual lines (N = 13), see *Figure 3—figure supplement 3D*. For (d-j), cells were treated with 0.25 μg/ml doxycycline for 2 days prior to and during the assay and fresh medium was added 4 hr prior to sample collection.

DOI: https://doi.org/10.7554/eLife.45572.009

The following source data and figure supplements are available for figure 3:

*Figure 3 continued on next page*

*Figure 3 continued*

**Source data 1.** CDO1 is preferentially silenced in KEAP1 mutant NSCLC and antagonizes proliferation.
DOI: https://doi.org/10.7554/eLife.45572.016
**Figure supplement 1.** CDO1 is epigenetically silenced in NSCLC.
DOI: https://doi.org/10.7554/eLife.45572.010
**Figure supplement 1—source data 1.** CDO1 is epigenetically silenced in NSCLC.
DOI: https://doi.org/10.7554/eLife.45572.011
**Figure supplement 2.** CDO1 protein does not correlate with mRNA expression.
DOI: https://doi.org/10.7554/eLife.45572.012
**Figure supplement 2—source data 1.** CDO1 protein does not correlate with mRNA expression.
DOI: https://doi.org/10.7554/eLife.45572.013
**Figure supplement 3.** NRF2 promotes CDO1 expression in NSCLC.
DOI: https://doi.org/10.7554/eLife.45572.014
**Figure supplement 3—source data 1.** NRF2 promotes CDO1 expression in NSCLC.
DOI: https://doi.org/10.7554/eLife.45572.015

restored NRF2 expression in combination with CDO1$^{WT}$, CDO1$^{Y157F}$, or GFP. NRF2 restoration in these cells led to higher expression of endogenous xCT and accumulation of ectopically expressed CDO1 compared to GFP control (*Figure 3F*). Next, we selected the two KEAP1$^{WT}$ NSCLC cell lines that had the lowest ectopic CDO1 accumulation and low intracellular CYS in our cell line panel, H1299 and H1975. Using a NRF2$^{T80K}$ mutant that is unable to bind KEAP1 (*Berger et al., 2017*), the effects of NRF2 on CDO1 accumulation were recapitulated in these KEAP1$^{WT}$ NSCLC cell lines (*Figure 3G*). Interestingly, we observed that NRF2$^{T80K}$ could also promote the accumulation of endogenous CDO1 in H1299 cells. Further, NRF2 stabilization with the ROS inducing agent β-lapachone promoted NRF2, CYS and CDO1 accumulation in H1299 and H1975 cells (*Figure 3—figure supplement 3A,B*), which was most pronounced the day following the 4 hr treatment window. Consistently, NRF2 depletion in KEAP1$^{MUT}$ NSCLC cells following KEAP1$^{WT}$ expression led to CDO1 depletion (*Figure 3H*), although the effects were more modest than what was observed with NRF2 activation. Notably, NRF2 expression promoted intracellular CYS accumulation, while NRF2 depletion impaired CYS accumulation (*Figure 3I*), supporting a role for intracellular CYS in CDO1 stabilization. To directly assess the requirement for CYS, A549 cells were cultured in high or low (CYS)$_2$ and CDO1 levels were found to be dependent on CYS availability (*Figure 3—figure supplement 3C*).

Next, we examined the consequence of CDO1 expression on cellular proliferation. Using the isogenic NRF2 KO A549 cell system, we observed that CDO1 expression significantly impaired the proliferation of NRF2-expressing cells, while no effect was observed on NRF2 KO cells (*Figure 3J*). Looking more broadly, we observed that CDO1 expression generally antagonized the proliferation of NSCLC cell lines and proliferation inhibition was strongly correlated with CDO1 protein expression, but not RNA expression (*Figure 3K* and *Figure 3—figure supplement 3D,E*). Overall, these results demonstrate that NRF2 and other mechanisms of intracellular CYS accumulation promote CDO1 accumulation, which leads to a selective growth disadvantage in lung cancer cells.

## CDO1 depletes CYS, leading to its export as CSA

To evaluate the mechanism by which CDO1 expression impaired proliferation we interrogated CYS metabolism following CDO1 expression. CYS has multiple intracellular fates, including the synthesis of GSH. CDO1 metabolizes CYS to CSA, which is then decarboxylated to HTAU (*Figure 4A*). CDO1$^{WT}$ and CDO1$^{Y157F}$-expressing A549 cells were fed fresh (CYS)$_2$-containing medium and sulfur containing metabolites were quantified over time (*Figure 4B–D*). CDO1$^{WT}$, but not the enzyme-inactive CDO1$^{Y157F}$, limited intracellular CYS levels and promoted the accumulation of CSA, which peaked at 4 hr (*Figure 4B*). We observed a steady increase in both CDO1$^{Y157F}$ and CDO1$^{WT}$ protein over the 24 hr time period, although CDO1$^{WT}$ protein levels were lower, consistent with the intracellular CYS levels (*Figure 4—figure supplement 1A*). Interestingly, unlike what was observed in MEFs, the levels of GSH, HTAU, and TAU were not changed over the time course of this assay (*Figure 4C*). We also interrogated metabolite changes in the medium and observed that (CYS)$_2$ was rapidly depleted from the medium by 24 hr, while CSA steadily accumulated (*Figure 4D*). Based on this time course, 4 hr was selected for all subsequent experiments to prevent (CYS)$_2$ starvation by

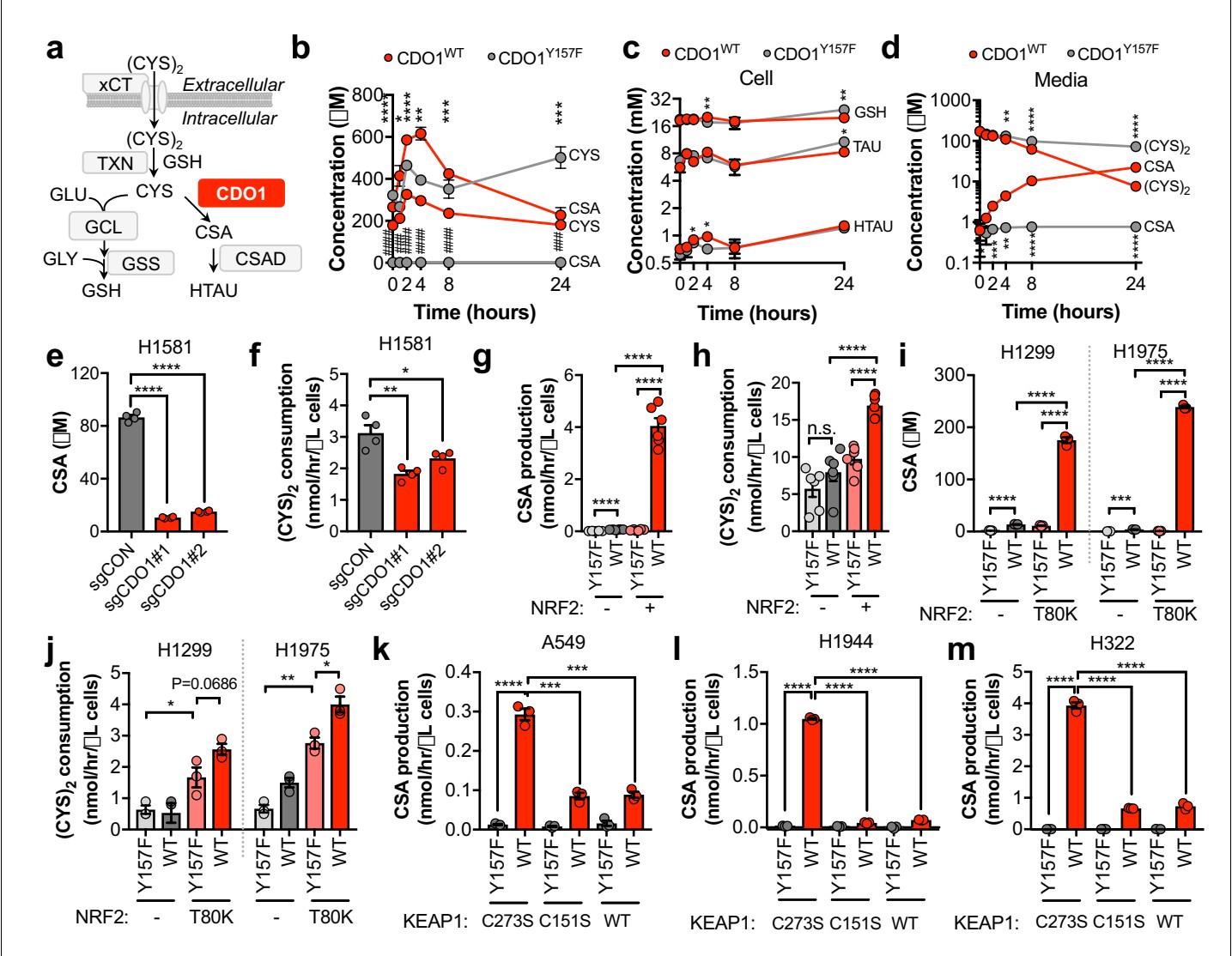

**Figure 4.** CDO1 depletes cyst(e)ine, leading to its export as CSA. (a) Schematic depicting intracellular cysteine metabolism. Following uptake of cystine via xCT, it is reduced to cysteine via the cellular antioxidant systems. Cysteine then enters glutathione synthesis mediated by GCL and GSS, or undergoes irreversible metabolism by CDO1 to CSA, and subsequently to HTAU by CSAD. (b) Time-dependent quantification (0, 1, 2, 4, 8, 24 hr) of intracellular cysteine (CYS) and cysteine sulfinic acid (CSA) concentrations following medium replenishment of CDO1$^{WT}$-expressing (red) and CDO1$^{Y157F}$-expressing (gray) A549 cells. *= CDO1$^{WT}$ vs. CDO1$^{Y157F}$ CYS, #= CDO1$^{WT}$ vs. CDO1$^{Y157F}$ CSA. N = 3 replicates/group, except 8 hr CYS for which N = 6 replicates/group. (c) Time-dependent quantification of GSH, HTAU, and TAU in the extracts from (b). N = 3 replicates/group. (d) Time-dependent quantification of the levels of (CYS)$_2$ and CSA in the medium from the cells from (b). N = 3 replicates/group. (e) Intracellular CSA concentration of H1581 cells following expression of control (sgCON) or CDO1-targeting sgRNAs (sgCDO1 #1 and #2) and Cas9. N = 4 replicates/group. (f) Quantification of (CYS)$_2$ consumption from the medium of the cells from (e). N = 4 replicates/group. (g) CSA production rate of NRF2 KO A549 cells reconstituted with either pLX317 empty (NRF2 -) or pLX317-NRF2 (NRF2 +), followed by expression of CDO1$^{Y157F}$ (Y157F) or CDO1$^{WT}$ (WT). N = 6 replicates/group. (h) Quantification of (CYS)$_2$ consumption from the medium of the cells from (g). (i) Intracellular CSA concentration in H1975 and H1299 cells expressing either pLX317 empty (-) or pLX317-NRF2$^{T80K}$ (NRF2$^{T80K}$), followed by expression of CDO1$^{Y157F}$ (Y157F) or CDO1$^{WT}$ (WT). N = 3 replicates/group. (j) Quantification of (CYS)$_2$ consumption from the medium of the cells from (i). (k–m) CSA production rate of A549 (k), H1944 (l) and H322 (m) cells following expression of CDO1$^{Y157F}$ (Y157F) or CDO1$^{WT}$ (WT), and reconstituted with inactive KEAP1 (C273S), super repressor KEAP1 (C151S) or wild-type KEAP1 (WT). N = 3 replicates/group. For (g–m) cells were treated with 0.25 µg/ml doxycycline for 2 days prior to and during the assay and fresh medium was added 4 hr prior to sample collection.

DOI: https://doi.org/10.7554/eLife.45572.017

The following source data and figure supplements are available for figure 4:

**Source data 1.** CDO1 depletes cyst(e)ine, leading to its export as CSA.
DOI: https://doi.org/10.7554/eLife.45572.019

*Figure 4 continued on next page*

*Figure 4 continued*

**Figure supplement 1.** CSA only accounts for a fraction of CDO1-dependent $(CYS)_2$ consumption.
DOI: https://doi.org/10.7554/eLife.45572.018
**Figure supplement 1—source data 1.** CSA only accounts for a fraction of CDO1-dependent $(CYS)_2$ consumption.
DOI: https://doi.org/10.7554/eLife.45572.020

CDO1. Importantly, deletion of endogenous CDO1 in H1581 cells reduced CSA production and $(CYS)_2$ consumption (*Figure 4E,F*). To examine the NRF2-dependence of these metabolite alterations, we utilized the NRF2$^{KO}$ cells, KEAP1$^{WT}$ cells, or KEAP1$^{MUT}$ cells from *Figure 3*. NRF2 promoted CSA accumulation and $(CYS)_2$ consumption following CDO1$^{WT}$ expression in NRF2 KO cells (*Figure 4G,H*). This effect was recapitulated by NRF2$^{T80K}$ expression in KEAP1$^{WT}$ cells (*Figure 4I,J*). Consistently, NRF2 depletion by KEAP1$^{WT}$ expression in KEAP1$^{MUT}$ cells inhibited the CDO1-dependent production of CSA (*Figure 4K–M*). Interestingly, KEAP1$^{WT}$ expression did not robustly affect CDO1 protein levels in H322 cells but significantly impaired CSA production by CDO1. Intracellular CYS can also influence CDO1 activity promoting its catalytic efficiency (*Dominy et al., 2008*), which may explain these results. Collectively, these results demonstrate that CDO1 expression promotes the production of CSA from CYS, leading to CSA accumulation both intracellularly and extracellularly, and enhanced $(CYS)_2$ consumption.

## CDO1 restoration in NSCLC cells promotes sulfite production, thereby depleting cystine via sulfitolysis

We found that CSA only accounted for a fraction of CDO1-dependent $(CYS)_2$ depletion (*Figure 4—figure supplement 1B*), suggesting that CSA is metabolized to an alternative product in NSCLC cell lines. To further characterize the consequence of CDO1 expression on CYS metabolism, we performed untargeted metabolomics and found significant accumulation of $SO_3^{2-}$ in CDO1-expressing cells (*Figure 5A*). Importantly, CSA can be transaminated by the cytosolic aspartate aminotransferase (GOT1) to produce β-sulfinyl pyruvate, which spontaneously decomposes to pyruvate and $SO_3^{2-}$ (*Singer and Kearney, 1956*) (*Figure 5B*). While HTAU and TAU are non-toxic molecules that have important physiological functions (*Aruoma et al., 1988*; *Hansen et al., 2010*; *Schaffer et al., 2000*; *Suzuki et al., 2002*), $SO_3^{2-}$ is toxic at high levels due to its cleavage of disulfide bonds in proteins and small molecules, including $(CYS)_2$ (*Clarke, 1932*). Thus, we hypothesized that in addition to accounting for the fate of CSA metabolism, $SO_3^{2-}$ production may also contribute to $(CYS)_2$ depletion through disulfide cleavage, also known as sulfitolysis.

We next performed a quantitative analysis of $SO_3^{2-}$ levels following CDO1 expression. CDO1$^{WT}$-, but not CDO1$^{Y157F}$-expressing A549s demonstrated rapid accumulation of extracellular $SO_3^{2-}$ over the 24 hr time course following media replenishment (*Figure 5C*). Further, the accumulation of the product of the sulfitolysis reaction, cysteine-S-sulfate (CYS-SO$_3^-$), was also observed in the medium of CDO1$^{WT}$-expressing cells (*Figure 5D*, *Figure 5—figure supplement 1A*). We observed that CYS-SO$_3^-$ appeared earlier than $SO_3^{2-}$, and stopped accumulating once $(CYS)_2$ levels were depleted, suggesting that $SO_3^{2-}$ reacted with $(CYS)_2$ in a rapid and complete manner. To test this possibility, we incubated either CSA or sodium sulfite (Na$_2$SO$_3$) with culture medium in the absence of cells, and observed rapid and robust conversion of $(CYS)_2$ to CYS-SO$_3^-$ by Na$_2$SO$_3$ (*Figure 5E*), but not CSA (*Figure 5—figure supplement 1B*), within 5 min. Interestingly, similar depletion kinetics in this experiment could also be observed by substituting $(CYS)_2$ with oxidized glutathione (GSSG) (*Figure 5—figure supplement 1C,D*), although intracellular and extracellular levels of GSSG in cell culture were much lower than $(CYS)_2$ (*Figure 5—figure supplement 1E,F*), suggesting it is not the major target of $SO_3^{2-}$. These results demonstrate that $SO_3^{2-}$ is generated downstream of CDO1 and rapidly reacts with $(CYS)_2$, thereby depleting $(CYS)_2$ from the culture media.

To evaluate whether $SO_3^{2-}$ production was a consequence of our CDO1 overexpression system, we transduced Keap1$^{WT}$ and Keap1$^{R554Q}$ MEFs with our inducible CDO1 vectors (*Figure 5—figure supplement 2A*). Consistent with its regulation by intracellular CYS, ectopic CDO1 expression was significantly higher in Keap1$^{R554Q}$ MEFs compared to Keap1$^{WT}$ MEFs. While CDO1 overexpression promoted the accumulation of intracellular CSA and HTAU, and the depletion of CYS and GSH (*Figure 5—figure supplement 2B–G*), we did not observe the production of $SO_3^{2-}$ or CYS-SO$_3^-$ in MEFs

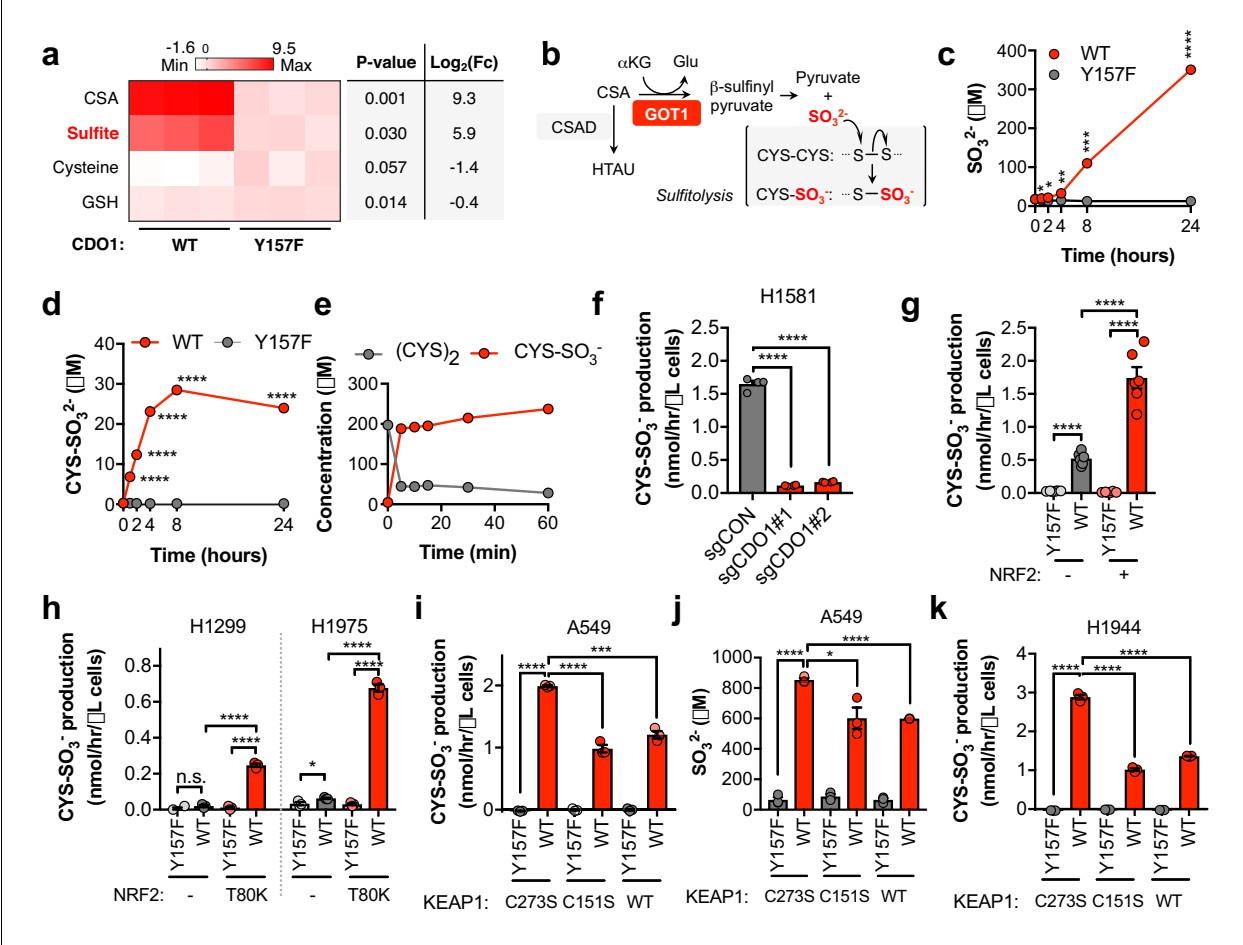

**Figure 5.** CDO1 expressing cells produce $SO_3^{2-}$, which further depletes $(CYS)_2$ via sulfitolysis. (**a**) Relative abundance of cysteine-related metabolites in cell extracts following $CDO1^{WT}$ or $CDO1^{Y157F}$ expression in A549 cells. Fresh medium was added 4 hr prior to extraction. Metabolites were analyzed by untargeted LC-MS. N.B. - cysteine and GSH were not derivatized with NEM for this analysis. N = 3 replicates/group. (**b**) GOT1 mediates the transamination of CSA to produce β-sulfinyl pyruvate, which decomposes to pyruvate and sulfite ($SO_3^{2-}$). Sulfite hydrolyzes $(CYS)_2$ to produce cysteine-S-sulfate ($CYS-SO_3^-$) in a process known as sulfitolysis. (**c,d**) Time-dependent quantification of the medium concentrations of and $SO_3^{2-}$ (**c**) and $CYS-SO_3^-$ (**d**) concentrations from *Figure 4d*. (**e**) Time-dependent quantification of $(CYS)_2$ and $CYS-SO_3^-$ following the addition of 1 mM $Na_2SO_3$ to RPMI +10% FBS in the absence of cells. N = 3 replicates/group. (**f**) $CYS-SO_3^-$ production rate of H1581 cells following expression of control (sgCON) or CDO1-targeting sgRNAs (sgCDO1 #1 and #2) and Cas9. N = 4 replicates/group. (**g**) $CYS-SO_3^-$ production rate of NRF2 KO A549 cells reconstituted with either pLX317 empty (NRF2 -) or pLX317-NRF2 (NRF2 +), followed by expression of $CDO1^{Y157F}$ (Y157F) or $CDO1^{WT}$ (WT). N = 6 replicates/group. (**h**) $CYS-SO_3^-$ production rate of H1975 and H1299 cells expressing either pLX317 empty (-) or pLX317-$NRF2^{T80K}$ ($NRF2^{T80K}$), followed by expression of $CDO1^{Y157F}$ (Y157F) or $CDO1^{WT}$ (WT). N = 3 replicates/group. (**i**) $CYS-SO_3^-$ production rate of A549 cells following expression of $CDO1^{Y157F}$ (Y157F) or $CDO1^{WT}$ (WT), and reconstituted with inactive KEAP1 (C273S), super repressor KEAP1 (C151S) or wild-type KEAP1 (WT). N = 3 replicates/group. (**j**) Intracellular $SO_3^{2-}$ concentration in the cells from (i). (**k**) $CYS-SO_3^-$ production rate of H1944 cells following expression of $CDO1^{Y157F}$ (Y157F) or $CDO1^{WT}$ (WT), and reconstituted with inactive KEAP1 (C273S), super repressor KEAP1 (C151S) or wild-type KEAP1 (WT). N = 3 replicates/group. For a, c, d, g-k, cells were treated with 0.25 µg/ml doxycycline for 2 days prior to and during the assay.

DOI: https://doi.org/10.7554/eLife.45572.021

The following source data and figure supplements are available for figure 5:

**Source data 1.** CDO1 expressing cells produce $SO_3^{2-}$, which further depletes $(CYS)_2$ via sulfitolysis.
DOI: https://doi.org/10.7554/eLife.45572.030
**Figure supplement 1.** Sulfite reacts with cystine and oxidized glutathione in the absence of cells.
DOI: https://doi.org/10.7554/eLife.45572.022
**Figure supplement 1—source data 1.** Sulfite reacts with cystine and oxidized glutathione in the absence of cells.
DOI: https://doi.org/10.7554/eLife.45572.023
**Figure supplement 2.** MEFs preferentially use the CSA decarboxylation pathway.
DOI: https://doi.org/10.7554/eLife.45572.024

*Figure 5 continued on next page*

*Figure 5 continued*

**Figure supplement 2—source data 1.** MEFs preferentially use the CSA decarboxylation pathway.

DOI: https://doi.org/10.7554/eLife.45572.025

**Figure supplement 3.** NRF2 promotes the CDO1-dependent production of sulfite in NSCLC cell lines.

DOI: https://doi.org/10.7554/eLife.45572.026

**Figure supplement 3—source data 1.** NRF2 promotes the CDO1-dependent production of sulfite in NSCLC cell lines.

DOI: https://doi.org/10.7554/eLife.45572.027

**Figure supplement 4.** CSA and sulfite are toxic to NSCLC cells.

DOI: https://doi.org/10.7554/eLife.45572.028

**Figure supplement 4—source data 1.** CSA and sulfite are toxic to NSCLC cells.

DOI: https://doi.org/10.7554/eLife.45572.029

(data not shown). Interestingly, MEFs express lower Got1 but higher Csad protein compared to A549 cells (*Figure 5—figure supplement 2A*), and NSCLC cell lines were uniformly low for CSAD and high for GOT1 (*Figure 5—figure supplement 3A*) suggesting that expression of CSA metabolic enzymes may be a key determining factor in the generation of HTAU vs. $SO_3^{2-}$. Importantly, deletion of endogenous CDO1 in H1581 cells resulted in a dramatic reduction in $CYS-SO_3^-$ production (*Figure 5F*), thereby demonstrating that $SO_3^{2-}$ is generated by CDO1 under physiological expression levels in lung cells.

To examine the NRF2-dependence of $(CYS)_2$ depletion via $SO_3^{2-}$, we utilized the NRF2 KO cells, KEAP1$^{WT}$, and KEAP1$^{MUT}$ cell lines. NRF2 promoted $CYS-SO_3^-$ production following CDO1$^{WT}$ expression (*Figure 5G*), which was accompanied by significant accumulation of both intracellular and extracellular $SO_3^{2-}$ (*Figure 5—figure supplement 3B,C*). These findings were recapitulated in KEAP1$^{WT}$ and KEAP1$^{MUT}$ NSCLC cell lines expressing NRF2$^{T80K}$ or following KEAP1 restoration, respectively (*Figure 5H–K*), with the exception of H322, which maintained $CYS-SO_3^{2-}$ production following KEAP1 restoration (*Figure 5—figure supplement 3G*). While KEAP1 restoration in these cells significantly reduced CSA production (*Figure 4M*), unlike A549 and H1944, intracellular CSA levels in H322 cells were still in the millimolar range (*Figure 5—figure supplement 3D–F*), suggesting that CSA transamination by GOT1 was saturated. Further, NRF2 stabilization with β-lapachone promoted CDO1-dependent CSA and $CYS-SO_3^-$ production and $(CYS)_2$ depletion in H1299 and H1975 cells (*Figure 5—figure supplement 3H–J*). Collectively, these results suggest that NRF2 induces CDO1-mediated sulfitolysis, thereby depleting extracellular $(CYS)_2$ in NSCLC cells.

Next, we examined the toxicity of CDO1 products to NSCLC cell lines. Treatment of cells with CSA and $Na_2SO_3$, but not HTAU, led to cytotoxicity (*Figure 5—figure supplement 4A*). We found that $(CYS)_2$ starvation and $Na_2SO_3$ treatment were universally toxic to NSCLC cell lines, which did not depend on NRF2 activity (*Figure 5—figure supplement 4B,C*). In addition, CDO1, CSA and $Na_2SO_3$ sensitized A549 cells to oxidative stress (*Figure 5—figure supplement 4D,E*), consistent with their ability to deplete $(CYS)_2$. Collectively, these results demonstrate that CSA and $SO_3^{2-}$ are toxic to NSCLC cells regardless of NRF2 activity, suggesting that resistance to $(CYS)_2$ starvation is not an inherent phenotype of NRF2$^{HIGH}$ cells. Rather, they are sensitive to CDO1 expression due to high intracellular CYS and CDO1 stabilization.

## Sulfitolysis is not required for the inhibition of proliferation by CDO1

To evaluate the role of GOT1 in CDO1-dependent sulfitolysis and cell growth inhibition, we generated GOT1 KO A549 and H460 cells (*Figure 6A*). Two independent clones of each were generated, and a sgRNA-resistant GOT1 cDNA was expressed in each to restore GOT1 expression (*Birsoy et al., 2015*). In support of GOT1 mediating the production of $SO_3^{2-}$ and $(CYS)_2$ depletion, GOT1 KO cells had significantly lower CDO1-dependent $(CYS)_2$ consumption and $SO_3^{2-}$ and $CYS-SO_3^-$ production rates compared to each parental line, which was rescued by GOT1 restoration (*Figure 6B*). Surprisingly, we observed that CDO1 antagonized cell proliferation independent of GOT1 expression and sulfitolysis, suggesting that the intracellular metabolism of CYS also contributes to the phenotype (*Figure 6C*). To address this mechanism, we evaluated whether CDO1 could limit CYS-dependent processes in NSCLC cell lines, similar to what was observed in MEFs. However, unlike in MEFs, CDO1 expression did not affect CYS utilization as the rates of GSH, Coenzyme A (CoA), and protein synthesis were similar (*Figure 6D–G*).

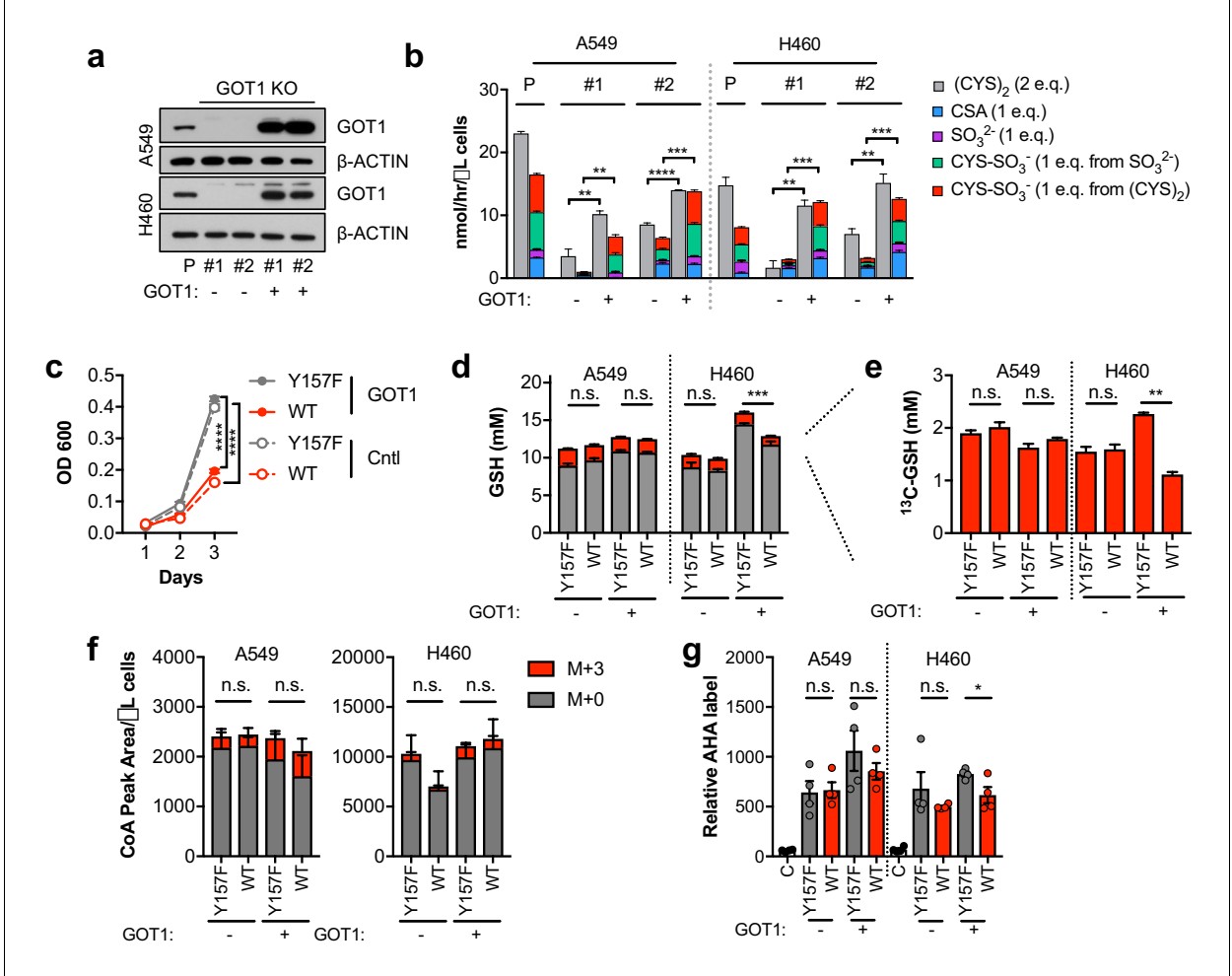

**Figure 6.** Sulfitolysis is not required for the inhibition of proliferation by CDO1. (a) Western blot analysis of GOT1 and β-ACTIN expression in parental (P) A549 and H460 cells, and GOT1 KO clones #1 and #2 for each cell line expressing empty pMXS (GOT1 -) and reconstituted with pMXS-GOT1 (GOT1 +). (b) Contribution of sulfitolysis to $(CYS)_2$ consumption by CDO1. CDO1-dependent $(CYS)_2$ consumption, and CSA, $SO_3^{2-}$, and $CYS-SO_3^-$ production were determined as cysteine molar equivalents. N = 3 replicates/group. (c) Analysis of the proliferation of GOT1 KO A549 cells from (a) expressing $CDO1^{Y157F}$ (Y157F) or $CDO1^{WT}$ (WT). Cells were collected on the indicated days, stained with crystal violet and their absorbance at 600 nm determined. N = 3 replicates/group. (d,e) Quantitation of glutathione (GSH) total ($^{12}C + {}^{13}C$) levels (d) and $^{13}C$-labeling only (e) from $^{13}C$-cystine in cells from (A) following expression of $CDO1^{Y157F}$ (Y157F) or $CDO1^{WT}$ (WT). $^{13}C$-label is shown in red, while $^{12}C$-label is gray. Cells were labeled for 1 hr. N = 3 replicates/group. (f) Analysis of Coenzyme A (CoA) labeling from $^{13}C$, $^{15}N$-cystine in cells from (a) expressing $CDO1^{Y157F}$ (Y157F) or $CDO1^{WT}$ (WT). $^{13}C,^{15}N$-label (M + 3) is shown in red, while $^{12}C$, $^{14}N$-label is gray. Cells were labeled for 4 hr. N = 3 replicates/group. (g) Analysis of protein synthesis rates with azidohomoalanine labeling in cells from (a) expressing $CDO1^{Y157F}$ (Y157F) or $CDO1^{WT}$ (WT). Cells treated with 50 µg/mL cyclohexamide (C) were used as a positive control for translation inhibition. N = 4 replicates/group. For **b–g,**) cells were treated with 0.25 µg/ml doxycycline for 2 days prior to and during the assay.

DOI: https://doi.org/10.7554/eLife.45572.031

The following source data is available for figure 6:

**Source data 1.** Sulfitolysis is not required for the inhibition of proliferation by CDO1.
DOI: https://doi.org/10.7554/eLife.45572.032

## CDO1 limits NADPH availability for cellular processes

Next, we examined other consequences of CDO1 expression in cells. After $(CYS)_2$ enters cells through its transporter xCT, it must be reduced to two CYS molecules using NADPH as the electron donor. As such, we hypothesized that the continual reduction of $(CYS)_2$ to CYS in CDO1-expressing cells would consume a significant amount of cellular NADPH. Indeed, we observed that the NADPH/$NADP^+$ ratio was lower following CDO1 expression in both GOT1 KO and GOT1 expressing cells

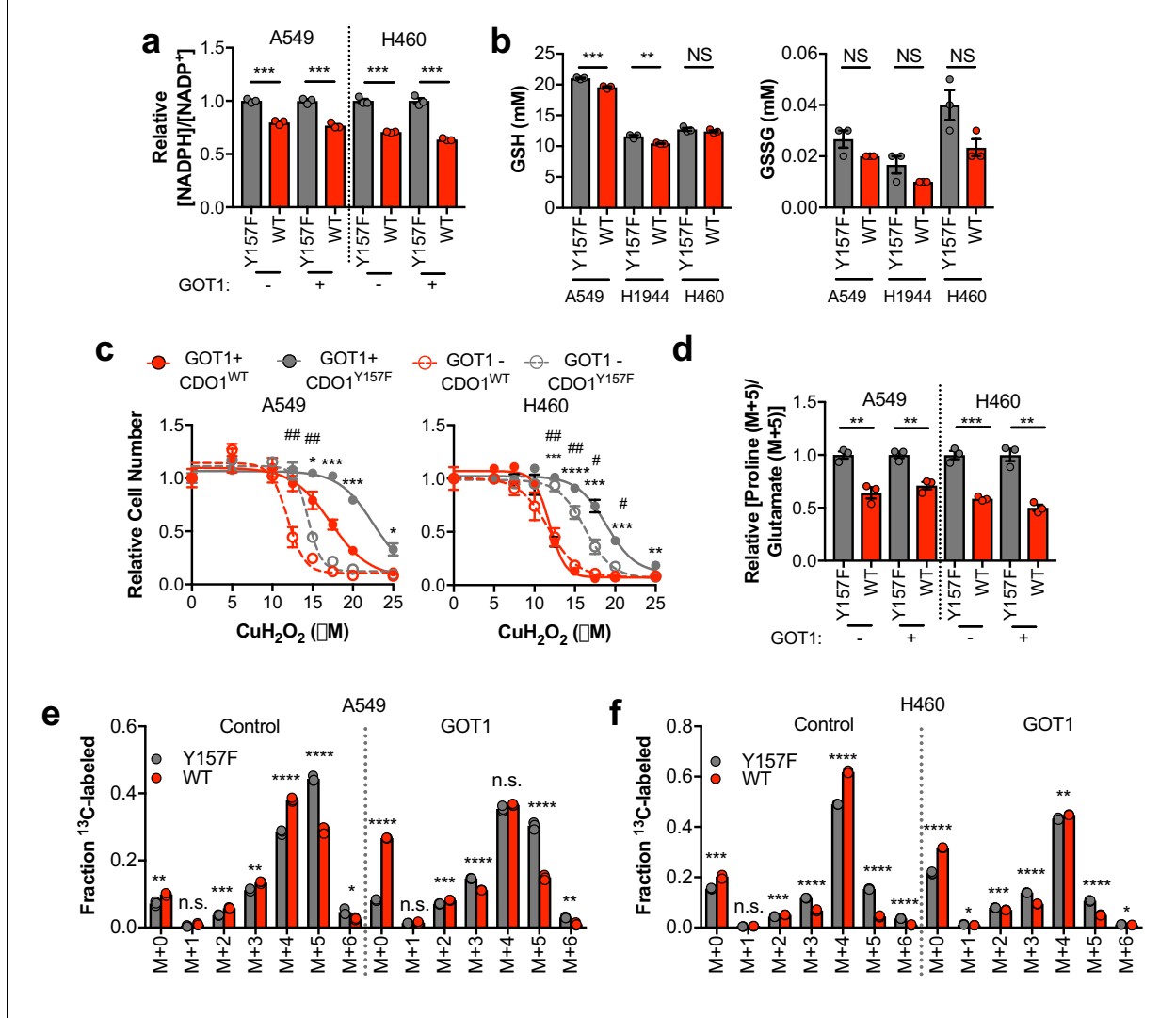

**Figure 7.** CDO1-dependent cystine reduction limits NADPH availability for cellular processes. (a) The NADPH/NADP + ratio was assayed following expression of CDO1[Y157F] (Y157F) or CDO1[WT] (WT) in A549 and H460 GOT1 KO cells expressing empty pMXS (GOT1 -) or reconstituted with pMXS-GOT1 (GOT1 +). N = 3 replicates/group. (b) Quantitation of reduced glutathione (GSH) and oxidized glutathione (GSSG) in KEAP1[C273S]-expressing cells from **Figure 4k,l** following expression of CDO1[Y157F] (Y157F) or CDO1[WT] (WT). N = 3 replicates/group. (c) A549 and H460 GOT1 KO cells expressing empty pMXS (dashed line, open circle) or reconstituted with pMXS-GOT1 (solid line, solid circle), followed by expression of CDO1[Y157F] (gray) or CDO1[WT] (red), were treated with 0–25 μM cumene hydroperoxide (CuH$_2$O$_2$) for 24 hr. Cell numbers were analyzed using crystal violet and normalized to untreated cells. *=GOT1+ CDO1[WT] vs. CDO1[Y157F], #=GOT1- CDO1[WT] vs. CDO1[Y157F]. N = 3 replicates/group. (d) Analysis of proline (M + 5) labeling from L-[U]-[13]C-glutamine in GOT1 KO A549 and H460 cells expressing empty pMXS (Control) or reconstituted with pMXS-GOT1 (GOT1), followed by expression of CDO1[Y157F] or CDO1[WT]. Proline M + 5 abundance was normalized to the abundance of its precursor glutamate M + 5, and then CDO1[WT] levels were normalized to CDO1[Y157F]. Cells were labeled for 1 hr in proline-free media. N = 3 replicates/group. (e,f) Mass isotopomer analysis of citrate labeling in GOT1 KO A549 (d) and H460 (e) cells expressing empty pMXS (Control) or reconstituted with pMXS-GOT1 (GOT1) following expression of CDO1[Y157F] or CDO1[WT] cultured with L-[U]-[13]C-glutamine for 4 hr. N = 3 replicates/group. For (a–f), cells were treated with 0.25 μg/ml doxycycline for 2 days prior to and during the assay.

DOI: https://doi.org/10.7554/eLife.45572.033

The following source data and figure supplements are available for figure 7:

**Source data 1.** CDO1-dependent cystine reduction limits NADPH availability for cellular processes.
DOI: https://doi.org/10.7554/eLife.45572.035

**Figure supplement 1.** Cystine uptake and reduction depletes NADPH.
DOI: https://doi.org/10.7554/eLife.45572.034

**Figure supplement 1—source data 1.** Cystine uptake and reduction depletes NADPH.
DOI: https://doi.org/10.7554/eLife.45572.036

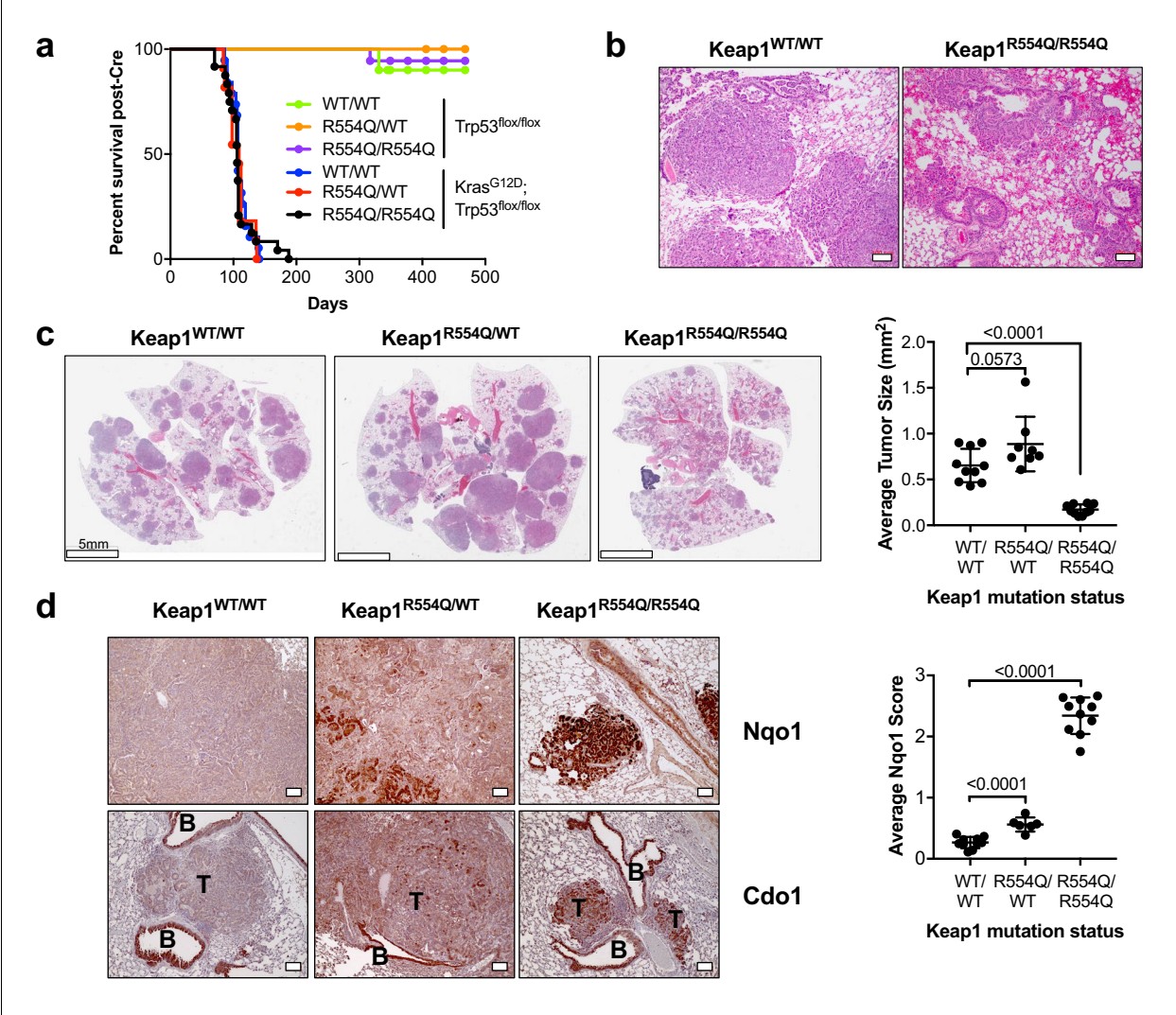

**Figure 8.** Nrf2 activation promotes Cdo1 expression in murine lung tumors in vivo. (**a**) Overall survival of *Trp53^flox/flox* and *Kras^G12D*; *Trp53^flox/flox* mice expressing homozygous *Keap1^WT* (WT/WT), heterozygous for *Keap1^R554Q* (R554Q/WT) or homozygous for *Keap1^R554Q* (R554Q/R554Q). (**b**) Representative hematoxylin and eosin (H&E) stained section depicting lung hemorrhage in *Kras^G12D*; *Trp53^flox/flox*; *Keap1^R554Q/R554Q* mice compared to *Kras^G12D*; *Trp53^flox/flox*; *Keap1^WT/WT* mice. (**c**) (Left) Representative H&E image of lung tumor burden in *Kras^G12D*; *Trp53^flox/flox* mice expressing *Keap1^WT/WT*, *Keap1^R554Q/WT* or homozygous for *Keap1^R554Q/R554Q*. (Right) Quantification of average lung tumor size per mouse. WT/WT (N = 10 mice), R554Q/WT (N = 8 mice), R554Q/R554Q (N = 10 mice). (**d**) (Left) Representative Nqo1 and Cdo1 immunohistochemistry of *Kras^G12D*; *Trp53^flox/flox* mouse tumors expressing *Keap1^WT/WT*, *Keap1^R554Q/WT* or homozygous for *Keap1^R554Q/R554Q*. B, Bronchiole; T, Tumor. (Right) Average score (scale = 0–4) for Nqo1 staining across mice. WT/WT (N = 10 mice), R554Q/WT (N = 6 mice), R554Q/R554Q (N = 10 mice).

DOI: https://doi.org/10.7554/eLife.45572.037

The following source data is available for figure 8:

**Source data 1.** Nrf2 activation promotes Cdo1 expression in murine lung tumors.
DOI: https://doi.org/10.7554/eLife.45572.038

(*Figure 7A*). Consistently, (CYS)$_2$ starvation had the opposite effect on the NADPH/NADP$^+$ ratio, which increased following starvation (*Figure 7—figure supplement 1A*). NADPH is critical for both antioxidant defense and cellular biosynthetic processes. While we found that the decrease in the NADPH/NADP$^+$ ratio did not dramatically influence the levels of GSH and GSSG in non-stressed cells (*Figure 7B*), expression of CDO1 increased sensitivity to the lipid peroxidation inducer cumene hydroperoxide (CuH$_2$O$_2$) independent of GOT1 expression (*Figure 7C*), and CDO1 deletion in H1581 cells promoted CuH$_2$O$_2$ resistance (*Figure 7—figure supplement 1B*). Next, we placed cells

into detached conditions, which has been shown to increase reliance on IDH1-dependent reductive carboxylation to promote NADPH generation in the mitochondria (*Jiang et al., 2016*). Consistently, CDO1 expression significantly impaired the ability of NSCLC cell lines to grow in soft agar (*Figure 7—figure supplement 1C*). Next, we examined the consequence of the altered $NADPH/NADP^+$ ratio on NADPH-dependent metabolic reactions. Using $^{13}C_5$-glutamine tracing, we observed that glutamine readily entered the TCA cycle to produce M + 4 citrate, which was unaffected or increased following CDO1 expression (*Figure 7E,F*), but CDO1 impaired both NADPH-dependent synthesis of proline from glutamate (*Figure 7D*), and NADPH-dependent reductive carboxylation of α-ketoglutarate to produce M + 5 citrate (*Figure 7E,F*). The antiproliferative effects of CDO1 were CYS-dependent, as either inhibition of $(CYS)_2$ uptake with erastin or low $(CYS)_2$ conditions resulted in loss of CDO1 expression and completely rescued the CDO1-induced proliferation defect (*Figure 7—figure supplement 1D–F*). Collectively, these results suggest that CDO1 further inhibits cellular processes by limiting NADPH availability, thereby impairing cellular proliferation.

## Nrf2 activation promotes Cdo1 accumulation in lung tumors in vivo

We next examined the ability of NRF2 to promote CDO1 stabilization under physiological $(CYS)_2$ conditions. To this end, we generated $Kras^{G12D}$; $Trp53^{flox/flox}$ lung tumor mice expressing either wild-type $Keap1$ ($Keap1^{WT}$), heterozygous for $Keap1^{R554Q}$ ($Keap1^{R554Q/WT}$), or homozygous for $Keap1^{R554Q}$ ($Keap1^{R554Q/R554Q}$). We chose this model because $Keap1$ deletion in the $Kras^{G12D}$; $Trp53^{flox/flox}$ lung tumor model was recently shown to activate Nrf2 and promote cystine uptake (*Romero et al., 2017*). We found that $Kras^{G12D}$; $Trp53^{flox/flox}$ mice had similar survival regardless of Keap1 mutation status (*Figure 8A*). However, survival was not related to tumor burden, as $Kras^{G12D}$; $Trp53^{flox/flox}$; $Keap1^{R554Q/R554Q}$ mice displayed significant lung hemorrhage at endpoint (*Figure 8B*) despite small tumors (*Figure 8C*). Quantification of average lung tumor size revealed that $Kras^{G12D}$; $Trp53^{flox/flox}$ mice expressing one copy of $Keap1^{R554Q}$ had modestly larger tumors, consistent with the findings of Romero et al., and in agreement with the dominant negative activity displayed by select Keap1 mutants (*Suzuki et al., 2011*). Surprisingly, complete loss of Keap1 function in the $Keap1^{R554Q/R554Q}$ mice led to significantly smaller tumors (*Figure 8C*). Further, Keap1 loss of function led to a gradient of Nrf2 activation with $Keap1^{WT}$ tumors expressing little to no Nqo1, $Keap1^{R554Q/+}$ tumors displayed increased Nqo1 expression, and $Keap1^{R554Q/R554Q}$ tumors were strongly positive (*Figure 8D*). Importantly, $Keap1^{R554Q/R554Q}$ tumors strongly expressed Cdo1 (*Figure 8D*), demonstrating that Nrf2 activation promotes Cdo1 accumulation under physiological conditions in vivo, and suggesting that Cdo1 may impede tumor progression. Additional work is needed to understand whether induction of Cdo1, or even Nrf2, is responsible for the apparent block in tumorigenesis observed in the $Keap1^{R554Q/R554Q}$ mice, or whether an alternative Keap1 substrate may mediate these effects.

## Discussion

The carbon, nitrogen and sulfur molecules of CYS are used for diverse cellular processes that are required for both homeostasis and proliferation. The carbon, nitrogen and sulfur atoms are incorporated into protein, GSH, TAU and CoA. Further, the sulfur atom of cysteine is incorporated into iron-sulfur (Fe-S) clusters. CYS is generally thought to be more limiting than glycine or glutamate for GSH synthesis in most tissues (*Stipanuk et al., 2006*). CDO1 plays a critical role in limiting CYS availability and toxicity (*Jurkowska et al., 2014*). CYS promotes both the stability and activity of CDO1 (*Stipanuk et al., 2009*), leading to the irreversible metabolism of CYS to CSA (*Stipanuk et al., 2009*). This mechanism of regulation prevents toxicity associated with CYS accumulation. However, the contribution of this regulatory process to CYS availability in cancer was not well understood.

Our findings implicate CDO1 as a metabolic liability for lung tumor cells with high intracellular CYS levels, particularly those with NRF2/KEAP1 mutations (*Figure 9*). We find that high intracellular CYS levels are a common feature of lung cancer cell lines, suggesting that NRF2-independent mechanisms exist to promote $(CYS)_2$/CYS uptake. Indeed, $(CYS)_2$ uptake is regulated by many signaling pathways, including EGFR, mTORC2 and p53 (*Gu et al., 2017*; *Jiang et al., 2015*; *Tsuchihashi et al., 2016*). Further, de novo CYS synthesis via the transsulfuration pathway (*Prigge et al., 2017*), direct CYS transport, or decreased CYS utilization may play a role in CYS accumulation. Notably, CDO1 promoter methylation is common across multiple cancer types

(*Brait et al., 2012*; *Jeschke et al., 2013*) raising the possibility that CDO1 antagonizes the proliferation or viability of other cancers through similar mechanisms. We find that CDO1 promotes the wasting of the carbon, nitrogen and sulfur molecules of CYS as CSA and $SO_3^{2-}$, promotes $(CYS)_2$ depletion by $SO_3^{2-}$, and induces depletion of NADPH. Any of these mechanisms could contribute to its growth suppressive function in vivo, but additional work is needed to evaluate the consequence of CDO1 loss in the relevant tumor microenvironment. Importantly, while limited GSSG is present in our cells and media, Sullivan et al. find that GSSG levels may exceed $(Cys)_2$ levels in murine tumor interstitial fluid (*Sullivan et al., 2019*), suggesting that GSSG may be a major target of $SO_3^{2-}$ in vivo. Our findings suggest that oncogene-induced metabolic processes can be unfavorable, and warrant further investigation into the selection against metabolic processes in the context of specific driving oncogenes or nutritional states.

A surprising finding from this study is that NRF2 stabilization promotes the accumulation of intracellular CYS to levels that far exceed those which are necessary for CYS-dependent metabolic processes in cancer cells. Normal intracellular CYS concentrations are approximately 100 µM (*Stipanuk et al., 2006*), around the Km for GCLC for CYS and typically an order of magnitude lower than GSH levels, consistent with our observed concentrations in wild-type MEFs. However, in many cases KEAP1 mutant NSCLC cell lines and even some KEAP1 wild-type lines accumulated CYS to millimolar levels without any apparent toxicity. CYS toxicity is a poorly described phenomenon that has been attributed to the reactivity of the free thiol on the CYS molecule, the production of hydrogen sulfide ($H_2S$), autooxidation and free radical formation, and other mechanisms (*Stipanuk et al., 2006*). More work is needed to understand whether high intracellular CYS levels are a vulnerability of CDO1-silenced cancer cells as a consequence of these mechanisms, or whether there is an advantage to high intracellular CYS.

We observe that Cdo1 modestly limits GSH synthesis in MEFs, similar to what has been demonstrated in the liver, where deletion of Cdo1 in vivo resulted in the accumulation of CYS and GSH (*Roman et al., 2013*). By contrast, CDO1 depleted intracellular CYS but did not limit GSH synthesis or other CYS-dependent processes in NSCLC cell lines. One potential explanation for this difference is that cancer cells are more efficient at maintaining intracellular CYS levels and/or GSH synthesis. While more efficient maintenance of intracellular CYS is an advantage for CYS-dependent metabolism, it is also a liability because considerable resources in the form of reducing power must be

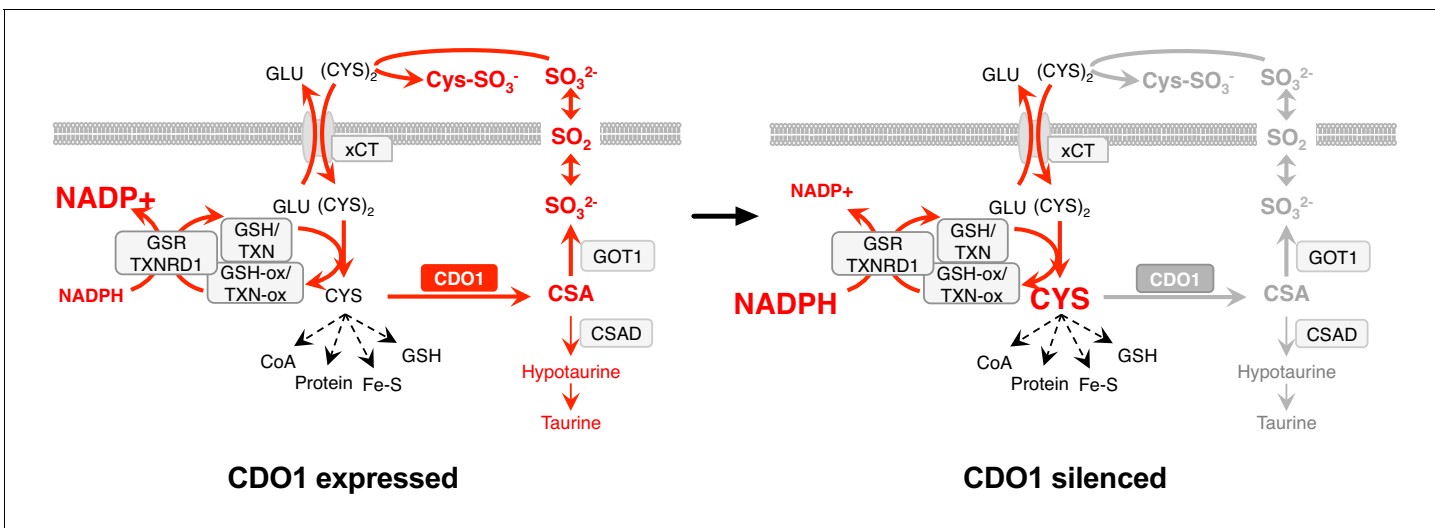

**Figure 9.** Model: CDO1 antagonizes the growth and survival of KEAP1[MUT] cells by producing toxic products and depleting NADPH. (Left) Elevated intracellular cysteine (CYS) stabilizes CDO1, leading to the production of CSA and $SO_3^{2-}$. In turn, $SO_3^{2-}$ depletes cystine via sulfitolysis to produce CYS-$SO_3^-$, further limiting cysteine availability and utilization. $SO_2$ (sulfur dioxide) exists in equilibrium with $SO_3^{2-}$ and is a gas that may diffuse in and out of cells. The continual reduction of $(CYS)_2$ to CYS depletes cellular NADPH, limiting its availability for cellular processes. (Right) CDO1 silencing promotes the accumulation of intracellular CYS and NADPH, and prevents wasting of CYS as CSA and toxic $SO_3^{2-}$. GSR, glutathione reductase. TXNRD1, thioredoxin reductase 1. Fe-S, iron sulfur cluster.
DOI: https://doi.org/10.7554/eLife.45572.039

committed to continually reduce $(CYS)_2$ to replenish the CYS pool. Although we do not know the rate of NADPH production in our cell lines, comparison with NADPH production rates in other lines (*Fan et al., 2014*) suggest that CDO1-dependent $(CYS)_2$ consumption would consume a significant fraction of the cellular NADPH produced, consistent with our observations.

Interestingly, we find that CSA is differentially metabolized in cancer cells and mouse embryonic fibroblasts, which correlates with differential expression of CSAD and GOT1. Partitioning between the decarboxylation and transamination reactions is likely influenced by the levels and activities of CSAD and GOT1, but this is not well studied. Evaluation of CSA metabolism at the organismal level has shown that decarboxylation of CSA via CSAD predominates over transamination by GOT1 (*Weinstein et al., 1988*) but little is known about this partitioning in individual tissues. Consistently, CSAD has a lower Km for CSA than GOT1 (*Recasens et al., 1980*; *Wu, 1982*). Importantly, HTAU and TAU are non-toxic molecules that have important physiological functions in controlling osmolarity, mitochondrial function, cellular redox, and other processes (*Aruoma et al., 1988*; *Hansen et al., 2010*; *Schaffer et al., 2000*; *Suzuki et al., 2002*). TAU biosynthesis is not a required process in non-hepatic tissues, however, as adequate TAU is supplied from the liver via the blood supply (*Stipanuk, 2004*). By contrast, $SO_3^{2-}$ is toxic in large quantities (*Menzel et al., 1986*) and thus excessive CSA transamination is likely disadvantageous in most cell types. Interestingly, hemorrhagic pulmonary edema is a consequence of sulfur dioxide exposure in humans (*Charan et al., 1979*), and additional work is needed to determine if Cdo1-mediated $SO_3^{2-}$ production is the cause of lung hemorrhage in the Keap1$^{R554Q/R554Q}$ mice. It is unclear what function the CSA transamination reaction serves for cells. Unlike the decarboxylation pathway, it retains carbon and nitrogen intracellularly as pyruvate and glutamate. Sulfite oxidase (SUOX) can reduce sulfite $(SO_3^{2-})$ to sulfate $(SO_4^{2-})$, which is an important precursor for sulfation-based detoxification of phenols, hydroxylamines, or alcohols to sulfate esters. Sulfation is also an important post-translational modification of proteins.

There are currently several approaches being developed to target aberrant CYS metabolism in cancer, including NRF2/KEAP1 mutant cancer. Despite their increased antioxidant capacity, KEAP1 mutant cells are still sensitive to $(CYS)_2$ starvation, which could be targeted with cyst(e)inase (*Cramer et al., 2017*). Further, $(CYS)_2$ uptake via xCT results in a central carbon imbalance and dependence on glutamine that can be targeted with glutaminase inhibitors (*Romero et al., 2017*). Our findings suggest excessive $(CYS)_2$ uptake can also impair NADPH-dependent processes and activating CDO1 expression may have unique therapeutic potential compared to $(CYS)_2$-depleting strategies. We find that Cdo1 accumulates in tumors from our Keap1$^{R554Q/R554Q}$ mutant lung cancer GEMM, which is correlated with a block in lung tumor formation. Further work is needed to determine whether Cdo1 impairs tumorigenesis in this model, and to evaluate the consequences of strategies to induce CDO1 expression on tumor growth and metabolism in vivo.

# Materials and methods

## Key resources table

| Reagent type (species) or resource | Designation | Source or reference | Identifiers | Additional information |
|---|---|---|---|---|
| Chemical compound, drug | Cycloheximide | Sigma Aldrich | 01810–1G | |
| Chemical compound, drug | Pyridine | Sigma Aldrich | 270970–1L | |
| Chemical compound, drug | CuCl$_2$ | Sigma Aldrich | 203149–10 | |
| Chemical compound, drug | Methoxylamine hydrochloride | Sigma Aldrich | 226904–5G | |
| Chemical compound, drug | Ethyl acetate (LC-MS grade) | Sigma Aldrich | 34972–1 L-R | |
| Chemical compound, drug | Methanol (HPLC grade) | Sigma Aldrich | 34860–1 L-R | |

*Continued on next page*

*Continued*

| Reagent type (species) or resource | Designation | Source or reference | Identifiers | Additional information |
|---|---|---|---|---|
| Chemical compound, drug | Cysteine | Sigma Aldrich | C6852-25G | |
| Chemical compound, drug | Cystine | Sigma Aldrich | C6727-25G | |
| Chemical compound, drug | L-Cysteine S-sulfate | Sigma Aldrich | C2196-25MG | |
| Chemical compound, drug | L-Cysteine sulfinic acid | Sigma Aldrich | 270881–1G | |
| Chemical compound, drug | Hypotaurine | Sigma Aldrich | H1384-100MG | |
| Chemical compound, drug | $Na_2SO_3$ | Sigma Aldrich | 71988–250G | |
| Chemical compound, drug | $[^{34}S]$-$Na_2SO_3$, | Sigma Aldrich | 753572 | |
| Chemical compound, drug | Doxycycline | Sigma Aldrich | D9891-1G | |
| Chemical compound, drug | $[^{13}C_2,^{15}N]$-Glutathione trifluoroacetate salt, | Sigma Aldrich | 683620 | |
| Chemical compound, drug | GSSG | Sigma Aldrich | G4376-250MG | |
| Chemical compound, drug | Decitabine (5-Aza-2'-deoxycytidine) | Sigma Aldrich | A3656-5MG | |
| Chemical compound, drug | Cumene hydroperoxide | Invitrogen | component of C10445 | |
| Chemical compound, drug | N-Ethylmaleimide | Chem-Impex International | 00142 | |
| Chemical compound, drug | $[^{13}C_5]$-glutamine, | Cambridge Isotope Labs | CLM-1822-H-0.25 | |
| Chemical compound, drug | $[^{13}C_3, ^{15}N]$-Cysteine | Cambridge Isotope Labs | CNLM-3871-H-0.25 | |
| Chemical compound, drug | $[^{13}C_3]$-Cysteine, | Cambridge Isotope Labs | CLM-4320-H-0.1 | |
| Chemical compound, drug | $[D_4]$-Cystine, | Cambridge Isotope Labs | DLM-1000–1 | |
| Chemical compound, drug | $[^{13}C_2]$-Taurine | Cambridge Isotope Labs | CLM-6622 | |
| Chemical compound, drug | $[D_5]$-Glutathione | Santa Cruz Biotechnology | sc-489493 | |
| Chemical compound, drug | $[D_4]$-Hypotaurine | CDN Isotopes | H1384-100MG | |
| Chemical compound, drug | Formic acid | Fisher Chemical | MFX04405 | |
| Chemical compound, drug | NaOH | Fisher Chemical | SS256500 | |
| Chemical compound, drug | MSTFA + 1% TMCS solution | Fisher Chemical | TS-48915 | |
| Chemical compound, drug | HPLC-grade water | Fisher Chemical | W5-1 | |
| Chemical compound, drug | Acetonitrile (HPLC grade) | Honeywell | 34967 | |
| Chemical compound, drug | Erastin | Cayman Chemical | 17754 | |

*Continued on next page*

Continued

| Reagent type (species) or resource | Designation | Source or reference | Identifiers | Additional information |
|---|---|---|---|---|
| Chemical compound, drug | β-lapachone | Dr. David Boothman | | |
| Chemical compound, drug | puromycin | Invivogen | ant-pr-1 | |
| Chemical compound, drug | blasticidin | Invivogen | ant-bl-1 | |
| Strain, strain background (mouse) | *LSL-Kras*$^{G12D}$ | (*Jackson et al., 2001*) | | |
| Strain, strain background (mouse) | *Trp53*$^{flox}$ | (*Marino et al., 2000*) | | |
| Strain, strain background (mouse) | Keap1R554Q | this paper | | A minigene containing a cDNA encoding wild -type exons 3–5, followed by a SV40 polyA signal, was inserted upstream of endogenous exon 3 of the *Keap1* gene. Codon 554 in endogenous exon four was mutated from arginine to glutamine |
| Recombinant DNA reagent | Keap1R554Q genotyping primers | this paper | Common (R) 5'-GCCACC CTATTCACAGACCA-3' Mutant (F) 5'-ATGGCCA CACTTTTCTGGAC 3' WT (F) 5'-GGGGGTAGA GGGAGGAGAAT-3' | WT PCR product = 326 bp Mutant PCR product = 584 bp |
| Recombinant DNA reagent | adenoviral-Cre | University of Iowa | VVC-U of Iowa-5 | |
| Recombinant DNA reagent | pRRL-CDO1 | this paper | | The LT3GEPIR vector backbone (*Fellmann et al., 2013*) was obtained from Johannes Zuber and the MiR-E cassette was excised to generate pRRL-GFP. GFP was excised and replaced with CDO1 using pQTEV-CDO1 (addgene# 31292) as a PCR template. |
| Recombinant DNA reagent | pRRL-CDO1 Y157F | this paper | | The enzyme inactive mutant (CDO1$^{Y157F}$) was generated from pRRL-CDO1 by site-directed mutagenesis of the wild-type protein. |
| Recombinant DNA reagent | pQTEV-CDO1 | Addgene | 31292 | |
| Recombinant DNA reagent | lentiCRISPR-V2 | (*Shalem et al., 2014*) | | |
| Recombinant DNA reagent | lentiCRISPR-V2 mCdo1 #2 | this paper | | progenitor: lentiCRISPR-V2; oligonucleotides for sgRNAs targeting mCDO1 (#2F 5'-caccgCGAGAGCAATCCCGCCGAGT-3', #2R 5'-aaacACTCGGCG GGATTGCTCTCGc-3') |
| Recombinant DNA reagent | lentiCRISPR-V2 mCdo1 #3 | this paper | | progenitor: lentiCRISPR-V2; oligonucleotides for sgRNAs targeting mCDO1 (#3F 5'-caccgCGAAGAGCTCATGTAA GATG-3', #3R 5'-aaacCATCTT ACATGAGCTCTTCGc-3') |

*Continued on next page*

*Continued*

| Reagent type (species) or resource | Designation | Source or reference | Identifiers | Additional information |
|---|---|---|---|---|
| Recombinant DNA reagent | lentiCRISPR-V2 hCDO1 #1 | this paper | | progenitor: lentiCRISPR-V2; oligonucleotides for sgRNAs targeting hCDO1 (F - 5'-caccgGAT GCGGATCAGATCAGCCA-3', R - 5'-aaacTGGCTGATCT GATCCGCATCc-3') |
| Recombinant DNA reagent | lentiCRISPR-V2 hCDO1 #2 | this paper | | progenitor: lentiCRISPR-V2; oligonucleotides for sgRNAs targeting hCDO1 (R - 5'-caccgCGAGAGCGACCCC ACCGAGT-3',R - 5'-aaacACTCG GTGGGGTCGCTCTCGc-3') |
| Recombinant DNA reagent | pLX317-NRF2 | Dr. Alice Berger, (*Berger et al., 2016*) | | |
| Recombinant DNA reagent | pLX317-NRF2$^{T80K}$ | Dr. Alice Berger, (*Berger et al., 2016*) | | |
| Recombinant DNA reagent | pLX317-empty | this paper | | pLX317-empty was generated from pLX317 -NRF2 by site directed mutagenesis. |
| Recombinant DNA reagent | pLenti KEAP1WT | this paper | | The KEAP1$^{WT}$ cDNA was provided by Dr. Christian Metallo (*Zhao et al., 2018*); pLenti-GFP-blast was generated from pLenti-GFP-puro (addgene #17448) |
| Recombinant DNA reagent | pLenti KEAP1C151S | this paper | | The KEAP1$^{C151S}$ cDNA was provided by Dr. Christian Metallo (*Zhao et al., 2018*); pLenti-GFP-blast was generated from pLenti-GFP -puro (addgene #17448) |
| Recombinant DNA reagent | pLenti KEAP1C273S | this paper | | The KEAP1$^{C273S}$ cDNA was provided by Dr. Christian Metallo (*Zhao et al., 2018*); pLenti-GFP-blast was generated from pLenti-GFP-puro (addgene #17448) |
| Recombinant DNA reagent | plentiCRISPR-sgGOT1 | addgene | 72874 | |
| Recombinant DNA reagent | pMXS-GOT1 | addgene | 72872 | |
| Recombinant DNA reagent | pMXS-empty | this paper | | pMXS-empty was generated from pMXS-GOT1 by site-directed mutagenesis. |
| Recombinant DNA reagent | pCMV-dR8.2 dvpr | addgene | 8455 | |
| Recombinant DNA reagent | pCMV-VSV-G | addgene | 8454 | |
| Cell line (Homo-sapiens) | Lenti-X 293T | Takara | 632180 | |
| Cell line (Homo-sapiens) | Phoenix-AMPHO | ATCC | CRL-3213 | RRID:CVCL_H716 |

*Continued on next page*

*Continued*

| Reagent type (species) or resource | Designation | Source or reference | Identifiers | Additional information |
| --- | --- | --- | --- | --- |
| Cell line (Homo-sapiens) | Calu3 | Dr John Minna, Hamon Cancer Center Collection (University of Texas-Southwestern Medical Center) | (*DeNicola et al., 2015*) | RRID:CVCL_0609 |
| Cell line (Homo-sapiens) | H1581 | Dr John Minna, Hamon Cancer Center Collection (University of Texas-Southwestern Medical Center) | (*DeNicola et al., 2015*) | RRID:CVCL_1479 |
| Cell line (Homo-sapiens) | H1975 | Dr John Minna, Hamon Cancer Center Collection (University of Texas-Southwestern Medical Center) | (*DeNicola et al., 2015*) | RRID:CVCL_1511 |
| Cell line (Homo-sapiens) | H2087 | Dr John Minna, Hamon Cancer Center Collection (University of Texas-Southwestern Medical Center) | (*DeNicola et al., 2015*) | RRID:CVCL_1524 |
| Cell line (Homo-sapiens) | H2347 | Dr John Minna, Hamon Cancer Center Collection (University of Texas-Southwestern Medical Center) | (*DeNicola et al., 2015*) | RRID:CVCL_1550 |
| Cell line (Homo-sapiens) | H1792 | Dr John Minna, Hamon Cancer Center Collection (University of Texas-Southwestern Medical Center) | (*DeNicola et al., 2015*) | RRID:CVCL_1495 |
| Cell line (Homo-sapiens) | H1944 | Dr John Minna, Hamon Cancer Center Collection (University of Texas-Southwestern Medical Center) | (*DeNicola et al., 2015*) | RRID:CVCL_1508 |
| Cell line (Homo-sapiens) | H322 | Dr John Minna, Hamon Cancer Center Collection (University of Texas-Southwestern Medical Center) | (*DeNicola et al., 2015*) | RRID:CVCL_1556 |
| Cell line (Homo-sapiens) | H460 | Dr John Minna, Hamon Cancer Center Collection (University of Texas-Southwestern Medical Center) | (*DeNicola et al., 2015*) | RRID:CVCL_0459 |
| Cell line (Homo-sapiens) | HCC15 | Dr John Minna, Hamon Cancer Center Collection (University of Texas-Southwestern Medical Center) | (*DeNicola et al., 2015*) | RRID:CVCL_2057 |
| Cell line (Homo-sapiens) | H2009 | ATCC | CRL-5911 | RRID:CVCL_1514 |
| Cell line (Homo-sapiens) | H1299 | ATCC | CRL-5803 | RRID:CVCL_0060 |
| Cell line (Homo-sapiens) | H1993 | ATCC | CRL-5909 | RRID:CVCL_1512 |
| Cell line (Homo-sapiens) | H441 | ATCC | HTB-174 | RRID:CVCL_1561 |

*Continued on next page*

Continued

| Reagent type (species) or resource | Designation | Source or reference | Identifiers | Additional information |
|---|---|---|---|---|
| Cell line (Homo-sapiens) | A549 | ATCC | CCL-185 | RRID:CVCL_0023 |
| Cell line (Homo-sapiens) | NRF2 KO A549 | Dr. Laureano de la Vega | (*Torrente et al., 2017*) | |
| Cell line (Mus musculus) | mouse embryonic fibroblasts | this paper | | MEFs were isolated from E13.5–14.5 day old embryos |
| Commercial assay or kit | JetPRIME | VWR | 89129–922 | |
| Commercial assay or kit | ImmPRESS HRP anti-rabbit kit | Vector Labs | MP-7451 | |
| Commercial assay or kit | E.Z.N.A. Total RNA Kit I | Omega Biotek | R6834-02 | |
| Commercial assay or kit | PrimeScript RT Master Mix | Takara | RR036A | |
| Commercial assay or kit | Taqman gene expression assays - mCdo1 | Thermo Fisher | 4448892, Mm00473573_m1 | |
| Commercial assay or kit | Taqman gene expression assays - hCDO1 | Thermo Fisher | 4448892, Hs01039954_m1 | |
| Commercial assay or kit | Taqman gene expression assays - mActb | Thermo Fisher | 4448892, Mm02619580_g1 | |
| Commercial assay or kit | Taqman gene expression assays - hACTB | Thermo Fisher | #4333762F | |
| Commercial assay or kit | CellTiter-Glo | Promega | G7571 | |
| Commercial assay or kit | DC protein assay | Biorad | 500112 | |
| Antibody | NRF2 (Rabbit mAb) | Cell Signaling Technologies | 12721 | RRID:AB_2715528; 1:1000 WB |
| Antibody | HSP90 (Rabbit pAb) | Cell Signaling Technologies | 4874 s | RRID:AB_2121214; 1:5000 WB |
| Antibody | TXN1 (Rabbit mAb) | Cell Signaling Technologies | 2429S | RRID:AB_2272594; 1:1000 WB |
| Antibody | α-tubulin (Mouse mAb) | Santa Cruz | sc-8035, clone TU-02 | RRID:AB_628408; 1:500 WB |
| Antibody | β-actin (Mouse mAb) | Thermo Fisher | AM4302, clone AC-15 | RRID:AB_2536382; 1:100,000 WB |
| Antibody | CDO1 (Rabbit pAb) | Abcam | ab53436 | RRID:AB_940958; 1:1000 WB, discontinued, verified with Sigma CDO1 antibody |
| Antibody | xCT (Rabbit pAb) | Abcam | ab37185 | RRID:AB_778944; 1:1000 WB |
| Antibody | GOT1 (Rabbit pAb) | Biovision | A1272 | RRID:AB_2801348; 1:1000 WB |
| Antibody | CSAD (Rabbit pAb) | LSBio | C375526 | RRID:AB_2801349; 1:1000 WB |
| Antibody | Anti-NQO1 antibody (Rabbit pAb) | Sigma Aldrich | HPA007308 | RRID:AB_1079501; 1:100, IHC; 1:1000 WB |

*Continued*

| Reagent type (species) or resource | Designation | Source or reference | Identifiers | Additional information |
|---|---|---|---|---|
| Antibody | Anti-CDO1 antibody (Rabbit pAb) | Sigma Aldrich | HPA057503 | RRID:AB_2683451; 1:100, IHC |
| Other | dialyzed FBS | Sigma Aldrich | F0392 | |
| Other | Cysteine/cystine, methionine and glutamine free RPMI | MP Biomedicals | 91646454 | |
| Other | Cysteine/cystine, methionine, pyruvate and glutamine free DMEM | Gibco | 21013024 | |
| Software, algorithm | MZmine 2 | (*Pluskal et al., 2010*) | Version 2.30 | |
| Software, algorithm | Thermo Xcaliber Qual Browser | Thermo Fisher | Version 4.0.27.19 | |
| Software, algorithm | El Maven | https://elucidatainc.github.io/ElMaven | Version 0.3.1 | |
| Software, algorithm | Agilent Mass Hunter Workstation Software - Qualitative Analysis | Agilent | Version B.07.00 | |
| Software, algorithm | Image Scope software | Aperio | | |

## Mice

The Keap1 targeting vector was constructed to contain homology arms and a minigene containing a cDNA encoding wild-type exons 3–5, followed by a SV40 polyA signal, inserted upstream of endogenous exon 3 of the *Keap1* gene. Codon 554 in endogenous exon four was mutated from arginine to glutamine in the targeting vector and the endogenous *Keap1* locus in C10 murine ES cells (*Beard et al., 2006*) was targeted and cells were selected with blasticidin. Positive clones were screened by copy number real-time PCR and injected into blastocysts. Mice were housed and bred in accordance with the ethical regulations and approval of the IACUC (protocol # R IS00003893). *Keap1^{R554Q}* mice were crossed with *LSL-Kras^{G12D}* and *Trp53^{flox}* mice for lung tumor studies. Lung tumor formation was induced by intranasal installation of $2.5 \times 10^7$ PFU adenoviral-Cre (University of Iowa) as described previously (*Jackson et al., 2001*). Viral infections were performed under isofluorane anesthesia, and every effort was made to minimize suffering.

## Immunohistochemistry and tumor volume analyses

Tissues were fixed in 10% formalin overnight before embedding in paraffin and sectioning. Sections were de-paraffinized in xylene and rehydrated in a graded alcohol series. Antigen retrieval was performed in 10 mM citrate buffer (pH 6.0) and endogenous peroxidase activity was quenched with 3% hydrogen peroxide. Immunohistochemical staining was performed with the ImmPRESS HRP anti-rabbit kit according to manufacturer's instructions (Vector Labs), followed by incubation with DAB substrate (Vector Labs). Staining intensity was graded on a 0 (no staining) – 4 (most staining) scale. Slides were scanned with the Aperio imager and tumor volume calculations were performed with Image Scope software (Aperio).

## Mouse embryonic fibroblast generation and culture

MEFs were isolated from E13.5–14.5 day old embryos and maintained in pyruvate-free DMEM (Corning) supplemented with 10% FBS. MEFs were infected with empty adenovirus or adenoviral-Cre (University of Iowa) at an MOI of 500 and used within four passages.

## Generation of lentivirus and retrovirus

For lentivirus production, Lenti-X 293 T cells (Clontech) were transfected at 90% confluence with Jet-PRIME (Polyplus). Packaging plasmids pCMV-dR8.2 dvpr (addgene # 8455) and pCMV-VSV-G (addgene #8454) were used. For retroviral production, Phoenix-AMPHO packaging cells (ATCC CRL-3213) were used.

## Lentiviral infection of MEFs

To generate CDO1 KO MEFs, MEFs were first infected with empty pLenti-CRISPR-V2, or pLenti-CRISPR-V2 encoding sgCDO1 #2 or #3 and selected with 1 µg/mL puromycin for 4 days, followed by infection with empty or cre-encoding adenovirus. To generate CDO1 overexpressing MEFs, MEFs were first infected with pRRL-CDO1 or GFP lentivirus and selected with 1 µg/mL puromycin for 4 days, followed by infection with empty or cre-encoding adenovirus, and then treated with doxycycline.

## NSCLC cell lines and culture

GOT1 knockout A549 and H460 cells were generated using the plentiCRISPR-sgGOT1 vector and KO clones were verified by western blotting. Cell lines were infected with lentivirus at a MOI <0.2 to achieve single copy integration and CDO1 expression verified by Q-RT-PCR. Cell lines were routinely tested and verified to be free of mycoplasma (MycoAlert Assay, Lonza). All lines were maintained in RPMI 1640 medium (Hyclone or Gibco) supplemented with 10% FBS without antibiotics at 37°C in a humidified atmosphere containing 5% $CO_2$ and 95% air.

## Analysis of *CDO1* mRNA expression and promotor methylation in patient samples

Patient lung adenocarcinoma data from The Cancer Genome Atlas (TCGA), with associated KEAP1 mutation and *CDO1* methylation status (Illumina HM450 Beadchip), was obtained via cBioPortal (*Cerami et al., 2012*; *Gao et al., 2013*). Patient normal lung and lung adenocarcinoma data from The Cancer Genome Atlas (TCGA), containing *CDO1* methylation status (Illumina HM450 Beadchip, beta value 0 [least methylated] – 1 [most methylated]) and mRNA expression data (RNA-seq RPKM), was obtained via the MethHC database (*Huang et al., 2015*). The *CDO1* promoter was defined as the region from 1.5 kb upstream to 0.5 kb downstream of the RefSeq TSS. The NRF2 activity score was determined as described previously (*DeNicola et al., 2015*).

## Analysis of mRNA expression

MEFs were seeded onto 6-well dishes and the cells were harvested at 70% confluence. NSCLC lines were harvested at 70% confluence, after treatment with 5 µM Decitabine for 3 days. During the treatment, medium was changed every 24 hr. For the NSCLC lines expressing CDO1, cells were pre-treated with DOX for 48 hr, and harvested at 4 hr after medium change. RNA was isolated with the E.Z.N.A. Total RNA Kit I (Omega Bio-Tek) according to the manufacturer's instructions. cDNA was synthesized from 500 ng of RNA using PrimeScript RT Master Mix (Takara) according to the manufacturer's instructions, and analyzed with Taqman gene expression assays.

## Preparation of NEM-derivatized cysteine and GSH internal standards

The N-ethylmaleamide (NEM) derivatized, isotope labeled, $[^{13}C_3, ^{15}N]$-cysteine-NEM, $[^{13}C_3]$-cysteine-NEM, $[^{13}C_2,^{15}N]$-GSH-NEM and $[D_5]$-GSH-NEM which were prepared by derivatizing the $[^{13}C_3, ^{15}N]$-cysteine, $[^{13}C_3]$-cysteine, $[^{13}C_2,^{15}N]$-GSH, and $[D_5]$-GSH standards with 50 mM NEM in 10 mM ammonium formate (pH = 7.0) at room temperature (30 min) as previously described (*Ortmayr et al., 2015*). $[^{13}C_4, ^{15}N_2]$-GSSG was prepared from the oxidation of $[^{13}C_2,^{15}N]$-GSH as described (*Zhu et al., 2008*).

## Generation of internal standards for the quantification of CSA, CYS-$SO_3^-$, and cystine

$[^{13}C_3, ^{15}N]$-CSA was synthesized from $[^{13}C_3, ^{15}N]$-cysteine as previously described (*Santhosh-Kumar et al., 1994*). Briefly, 1.23 mg of $[^{13}C_3, ^{15}N]$-cysteine was combined with 300 µL of 0.1 N NaOH, followed by addition of 300 nmol of $CuCl_2$. After a 12 hr incubation at 37°C, 400 µL of 0.1 N

NaOH was added and the product purified by solid phase extraction (SPE) as following. The reaction mixture was passed through 100 mg of AG MP-1M strong anion exchange resin (Bio-Rad), which was pre-equilibrated with 1.5 mL of 0.1N NaOH. After washing with 12 mL of HPLC-grade water, the isotope labeled $[^{13}C_3, ^{15}N]$-CSA was eluted with 1.8 mL of 1N HCl (*Figure 1—figure supplement 1B*). In addition to $[^{13}C_3, ^{15}N]$-CSA, $[^{13}C_6, ^{15}N_2]$-Cystine and $[^{13}C_3, ^{15}N]$-Cysteine S-sulfate (CYS-SO$_3^-$) (*Figure 5—figure supplement 1A*) were also obtained during $[^{13}C_3, ^{15}N]$-CSA synthesis as byproducts. The quantities of synthesized metabolites were determined using unlabeled CSA, CYS-SO$_3^-$, and Cystine standards and LC-MS/MS analysis as described (*Bennett et al., 2008*). Following quantification, these internal standards were used to quantify concentrations of CSA, CYS-SO$_3^-$ and cystine in cell and medium extracts.

## Sample preparation for non-targeted metabolite profiling

Cells were grown in 6-well dishes, quickly washed in cold PBS, and extracted in 80% methanol (0.5 mL for NSCLC cells, 0.2 mL for MEFs, −80°C). The extracts were cleared by centrifugation, and the metabolites in the supernatant were directly analyzed (MEFs) or dried by centrifugation under vacuum (NSCLC cells, SpeedVac, Thermo Scientific). The dried pellets were re-dissolved in 20 μL of HPLC grade water and analyzed by liquid chromatography-high resolution mass spectrometry (LC-HRMS). The extracellular metabolites from 10 μL of cell culture medium were extracted with 40 μL 100% methanol (−80°C). The extract was cleared by centrifugation, and 5 μL of the supernatant analyzed by LC-HRMS.

## Sample preparation for analysis of NEM-cysteine, NEM-GSH, cystine, GSSG, hypotaurine, and taurine

Cells were plated the day before extraction in 6-well dishes so they were 70% confluent at extraction. The medium was changed the next day and the cells harvested at the indicated time point. The metabolites were extracted and derivatized with 0.5 mL of ice-cold extraction solvent (80% MeOH:20% H$_2$O containing 25 mM NEM and 10 mM ammonium formate, pH 7.0) containing 20 μM $[^{13}C_3, ^{15}N]$-cysteine-NEM, 36.4 μM $[^{13}C_2, ^{15}N]$-GSH-NEM, 0.13 μM $[^{13}C_4, ^{15}N_2]$-GSSG, 10 μM $[D_4]$-Cystine, 20 μM $[^{13}C_2]$-Taurine, and 20 μM $[D_4]$-Hypotaurine, followed by incubation on ice for 30 min. The NEM-derivatized metabolite extracts were cleared by centrifugation and analyzed by LC-MS/MS via multiple reaction monitoring (LC-MRM) or LC-HRMS. Cell volumes and number were determined using a Scepter 2.0 cell counter (Millipore) and used to calculate the intracellular metabolite concentrations and to normalize metabolite consumption/production rates.

## Sample preparation for intracellular CSA quantification

The cell culture medium was aspirated after reserving 1 mL for extracellular metabolite analysis. Cells were quickly washed with ice cold PBS, and intracellular CSA was extracted with 80% MeOH including approximately 1 μM $[^{13}C_3, ^{15}N]$-CSA as an internal standard. The exact quantity was determined for each experiment using unlabeled CSA standard. After incubation for 15 min (−80°C) and scraping, the cell and metabolite mixture were transferred into a 1.5 mL tube. After clarification by centrifugation, the metabolite extracts were dried under vacuum and re-dissolved into HPLC grade water (20 μL for 6-well dishes or 40 μL for 100 mm dishes) for analysis by targeted LC-MRM.

## Sample preparation for extracellular CSA, CYS-SO$_3^-$, and cystine quantification

10 μL of cell culture medium was extracted at −80°C for at least 15 min with 40 μL of 100% ice cold MeOH containing approximately 25 μM $[^{13}C_3, ^{15}N]$-CSA, 10 μM $[^{13}C_3, ^{15}N]$-CYS-SO$_3^-$, and 300 μM $[^{13}C_6, ^{15}N_2]$-Cystine internal standards. The exact quantity was determined for each experiment using unlabeled standards. Following centrifugation (16,000 g, 20 min, 4°C), the extracellular metabolite extracts were transferred into a vial and analyzed by LC-MRM-based quantification.

## Sample preparation for sulfite quantification

A $[^{34}S]$-Na$_2$SO$_3$ standard was purchased from Sigma Aldrich (99% purity). The cells were extracted in 80% methanol containing 1 μM of $[^{34}S]$-Na$_2$SO$_3$ (0.5 mL for 6-well dish or 3 mL for 100 mm dishes, −80°C). The extracts were cleared by centrifugation, and the metabolites in the supernatant were

dried by centrifugation under vacuum. The pellets were re-dissolved in HPLC grade water (20 μL for 6-well dishes or 40 μL for 100 mm dishes) and analyzed by UPLC-Q-Exactive-HF (Thermo Fisher Scientific, Waltham, MA). The extracellular metabolites from 10 μL of cell culture medium were extracted with 40 μL 100% methanol containing 96.5 μM [$^{34}$S]-Na$_2$SO$_3$. The extract was cleared by centrifugation, and the metabolites in the supernatant were analyzed by LC-HRMS.

## [$^{13}$C$_6$]-Cystine tracing into cysteine, cystine, GSH, and hypotaurine

Medium containing 200 μM [$^{13}$C$_6$]-cystine were prepared as follows: For NSCLC cell line studies, cysteine/cystine, methionine and glutamine free RPMI (MP Biomedicals) was supplemented with 400 μM [$^{13}$C$_3$]-cysteine, 100 μM methionine, 2 mM glutamine and 10% dialyzed FBS (dFBS). For MEF studies, cysteine/cystine, methionine, pyruvate and glutamine free DMEM (Gibco) was supplemented with 400 μM [$^{13}$C$_3$]-cysteine, 200 μM methionine, 4 mM glutamine and 10% dFBS. Medium was sterile filtered using a 0.2 um PES vacuum filter and incubated for 48 hr at 4°C to allow oxidation of cysteine to cystine. Prior to $^{13}$C-labeling, cells were preconditioned in medium containing dFBS and $^{12}$C-cystine for 24 hr. Cells were labeled with $^{13}$C-cystine for the indicated timepoints, and then harvested. The metabolites were extracted and derivatized with NEM in ice-cold extraction solvent (80% MeOH:20% H$_2$O containing 25 mM NEM and 10 mM ammonium formate, pH = 7.0) which includes stable isotope labeled internal standards of 20 μM [$^{13}$C$_3$, $^{15}$N]-cysteine-NEM, 6.4 μM [D$_5$]-GSH-NEM, 10 μM [D$_4$]-Cystine, and 20 μM [D$_4$]-Hypotaurine followed by incubation on 4°C for 30 min. The NEM-derivatized metabolite extracts were cleared by centrifugation and analyzed by targeted LC-MRM.

## [$^{13}$C$_6$, $^{15}$N$_2$]-Cystine tracing into CoA

Cysteine/cystine, methionine and glutamine free RPMI (MP Biomedicals) was supplemented with 200 μM [$^{13}$C$_6$, $^{15}$N$_2$]-Cystine, 100 μM methionine, 2 mM glutamine and 10% dFBS as described above. NSCLC cell lines were preconditioned in medium containing dFBS and $^{12}$C-cystine for 24 hr in 10 cm dishes, followed by feeding with [$^{13}$C$_6$, $^{15}$N$_2$]-Cystine containing media. After 4 hr, the medium was aspirated, cells were quickly washed with ice cold PBS, and cellular metabolites extracted with 1 mL of 80% MeOH (−80°C, 15 min). After scraping, the metabolite extract was transferred into an Eppendorf tube and cleared by centrifugation (17000 g, 20 min, 4°C). The supernatant was dried under vacuum overnight, and stored at −80°C. Dried sample pellets were resuspended in HPLC-grade water (20 μl) and centrifuged at 20,000 g for 2 min to remove insoluble material. Supernatants (5 μl) were injected and analyzed using a hybrid 6500 QTRAP triple quadrupole mass spectrometer (AB/SCIEX) coupled to a Prominence UFLC HPLC system (Shimadzu) via selected reaction monitoring (MRM). ESI voltage was +4900V in positive ion mode with a dwell time of 3 ms per SRM transition. Approximately 10–14 data points were acquired per detected metabolite. The following SRM transitions were used: native Coenzyme A [M + 0] (Q1 = 768.13, Q2 = 261.13, collision energy =+ 39V) and $^{13}$C$_2$,$^{15}$N$_1$ isotope labeled Coenzyme A [M + 3] (Q1 = 771.13, Q2 = 264.13, collision energy = + 39V). Incorporation of $^{13}$C$_3$,$^{15}$N$_1$-cysteine into Coenzyme A yields $^{13}$C$_2$,$^{15}$N$_1$-Coenzyme A due to a decarboxylation step during Coenzyme A synthesis. Samples were delivered to the mass spectrometer via hydrophilic interaction chromatography (HILIC) using a 4.6 mm i.d x 10 cm Amide XBridge column (Waters) at 400 μl/min. Gradients were run starting from 85% buffer B (HPLC grade acetonitrile) to 42% B from 0 to 5 min; 42% B to 0% B from 5 to 16 min; 0% B was held from 16 to 24 min; 0% B to 85% B from 24 to 25 min; 85% B was held for 7 min to re-equilibrate the column. Buffer A was comprised of 20 mM ammonium hydroxide/20 mM ammonium acetate (pH = 9.0) in 95:5 water: acetonitrile. Peak areas from the total ion current for each metabolite SRM transition were integrated using MultiQuant v3.0 software (AB/SCIEX).

## [$^{13}$C$_5$]-glutamine tracing into citrate and proline

Cysteine/cystine, methionine, pyruvate and glutamine free DMEM (Gibco) was supplemented with 4mM [$^{13}$C$_5$]-glutamine, 200μM methionine, 200μM cystine and 10% dFBS. NSCLC cell lines were seeded onto 6-well dishes, preconditioned in DMEM containing $^{12}$C-glutamine and 10% dFBS for 24 hr, and subsequently the medium was changed to [$^{13}$C$_5$]-glutamine containing medium for the indicated time. The medium was aspirated, cells were quickly washed with ice cold PBS, and cellular metabolites were extracted with 500 μL of 80% MeOH (-80°C, 15 min). After scraping, the

metabolite extract was cleared by centrifugation (17000g, 20 min, 4°C). For [$^{13}$C$_5$]-glutamine → 13 (M+5) proline tracing, the supernatant was directly analyzed by LC-MS. For the [$^{13}$C$_5$]-glutamine → 13 citrate tracing, the supernatant was dried under vacuum and derivatization was performed for GC-MS analysis (*Carey et al., 2015*). The dried pellet was derivatized by adding 50 µL of methoxyl-amine hydrochloride (40 mg/mL in Pyridine) at 30°C for 90 min. After mixing the derivatized solution with 70 µL of Ethyl-Acetate in a glass vial, the metabolite mixture was further derivatized by adding 80 µL of MSTFA + 1%TMCS solution (37°C for 30 min). The final derivatized solution was transferred into a glass vial insert, and analyzed using an Agilent 7890B/5977B MSD (Agilent, Santa Clara, CA) GC-MS system.

## LC-HRMS conditions for NSCLC non-targeted metabolomics and sulfite quantification

Non-targeted metabolite profiling and sulfite quantification was conducted using Ultimate 3000 or Vanquish UPLC systems coupled to a Q Exactive HF (QE-HF) mass spectrometer equipped with HESI (Thermo Fisher Scientific, Waltham, MA). The analytical condition was as following. A Luna 3 µm HILIC 200 Å, LC Column 100 × 2 mm (Phenomenex, Torrance, CA, Part N: 00D-4449-B0) was applied as a stationary phase. The mobile phase A was 0.1 % formic acid in water, and the mobile phase B was 100% Acetonitrile (ACN). The column temperature was set to 30°C, and the gradient elution was conducted as following at a flow rate of 0.35 mL/min: 0 to 3 min, 0 % of phase A; 3 to 13 min, linear gradient from 0% to 80% of phase A; 13 to 16 min, 80% of phase A. The MS scan was operated in negative mode and the mass scan range was 58 to 870 m/z. The FT resolution was 120,000, and the AGC target was 3 × 10$^6$. The capillary temperature was 320°C, and the capillary voltage was 3.5 kV. The injection volume was 5µL. The MS peak extraction, the chromatographic peak extraction and deconvolution, the peak alignment between samples, gap filling, putative peak identification, and peak table exportation for untargeted metabolomics were conducted with MZmine 2 (2.23 Version) (*Pluskal et al., 2010*). The identity of CSA, cysteine, GSH, and sulfite were further confirmed by authentic standards. For sulfite quantification, [$^{34}$S]-HSO$_3^-$ and [$^{32}$S]-HSO$_3^-$ peaks were extracted with a 20 ppm MS filter, and their peak areas were manually integrated using Thermo Xcaliber Qual Browser. The quantification was based on previous methods (*Bennett et al., 2008*).

## LC-HRMS conditions for MEF non-targeted metabolomics and intracellular cysteine quantification

The LC-HRMS method is modified from previous study (*Cantor et al., 2017*) using Vanquish UPLC systems coupled to a Q Exactive HF (QE-HF) mass spectrometer equipped with HESI (Thermo Fisher Scientific, Waltham, MA). For chromatographic separation, a SeQuant ZIC-pHILIC LC column, 5 µm, 150 × 4.6 mm (MilliporeSigma, Burlington, MA) with a SeQuant ZIC-pHILIC guard column, 20 × 4.6 mm (MilliporeSigma, Burlington, MA) was used. The mobile phase A was 10mM ammonium carbonate and 0.05% ammonium hydroxide in water, and the mobile phase B was 100% Acetonitrile (ACN). The column temperature was set to 30°C, and the gradient elution was conducted as following at a flow rate of 0.25 mL/min: 0 to 13 min, linear gradient from 80% to 20% of phase B; 13 to 15 min, 20% of phase B. The MS1 scan was operated in both positive and negative mode for non-targeted metabolomics, or only in positive mode for the NEM-derivatized cysteine quantification. The mass scan range was 60 to 900 m/z. The MS resolution was 120,000, and the AGC target was 3 × 10$^6$. The capillary temperature was 320°C, and the capillary voltage was 3.5 kV. The injection volume was 5µL for both positive mode and negative mode. For the non-targeted metabolomics, the data analysis was performed with El Maven v0.3.1 (https://elucidatainc.github.io/ElMaven/) (*Clasquin et al., 2012*). Metabolite identification was based on comparison of both retention time and m/z value of sample peaks with an internal library (MSMLS Library, Sigma Aldrich). For cysteine quantification, the Cysteine-NEM and [$^{13}$C$_3$, $^{15}$N]-Cysteine-NEM peaks were extracted with a 10 ppm MS tolerance, and their peak areas were manually integrated using Thermo Xcaliber Qual Browser. The quantification was based on previous method (*Bennett et al., 2008*).

## LC-MRM conditions for targeted analyses

The targeted analysis of Cysteine-NEM, GSH-NEM, GSSG, cystine, hypotaurine, taurine, CSA, CYS-SO$_3^-$, GS-SO$_3^-$, Glutamic acid, and Proline were conducted by selected reaction monitoring (MRM) using an Ultimate 3000 UPLC system coupled to a Thermo Finnigan TSQ Quantum equipped with HESI (Thermo Fisher Scientific, Waltham, MA). The chromatographic separation was performed as previously described (*Liu et al., 2014*). As a stationary phase, an XBridge Amide Column 3.5μm (2.1 × 100mm) (Waters, Milford, MA) was used. The mobile phase A was 97% Water and 3% ACN (20 mM NH$_4$Ac, 15 mM NH$_4$OH, pH = 9.0) and the mobile phase B was 100% ACN. The column temperature was set to 40°C, and the gradient elution was as following at 0.35 mL/mL of flow rate: 0 to 3 min, linear gradient from 15% to 70% of Phase A; 3 to 12 min: linear gradient from 70% to 98% of Phase A; 12 to 15 min, sustaining 98% of Phase A. The MS acquisition operated at the positive or negative mode. The capillary temperature was 305 °C, the vaporizer temperature was 200 °C. The sheath gas flow was 75 and the auxiliary gas flow was 10. The spray voltage was 3.7 kV. The MRM conditions (parent ion → fragment ion; collision energy) of metabolites were as follows. Positive mode: Hypotaurine (m/z 110 → m/z 92; 10); [$^{13}$C$_2$]-Hypotaurine (m/z 112 → m/z 94; 10); [D$_4$]-Hypotaurine (m/z 114 → m/z 96; 10); Taurine (m/z 126 → m/z 108; 11); [$^{13}$C$_2$]-Taurine – (m/z 128 → m/z 110; 11); Cysteine-NEM (m/z 247 → m/z 158; 30); [$^{13}$C$_3$]-Cysteine-NEM (m/z 250 → m/z 158; 30); [$^{13}$C$_3$, $^{15}$N]-Cysteine-NEM (m/z 251 → m/z 158; 30); GSH–NEM m/z (m/z 433 → m/z 304; 15); [$^{13}$C$_2$,$^{15}$N]-GSH-NEM (m/z 436 → 307m/z; 15); [D$_5$]-GSH–NEM (m/z 438 → m/z 304; 15); GSSG (m/z 613 → m/z 355; 25), [$^{13}$C$_4$, $^{15}$N$_2$]-GSSG (m/z 619 → m/z 361; 25); Cystine (m/z 241→ m/z 74; 30); [D$_4$]-Cystine (m/z 245 → m/z 76; 30); [$^{13}$C$_6$]-Cystine (m/z 247 → m/z 76; 30); [$^{13}$C$_6$, $^{15}$N$_2$]-Cystine (m/z 249 → m/z 77; 30); CYS-SO$_3^-$ (m/z 202 → m/z 74; 27), [$^{13}$C$_3$, $^{15}$N]- CYS-SO$_3^-$ (m/z 206 → 77; 27); [M+5]-Glutamate (m/z 153 → m/z 88; 20); [M+5]-Proline (m/z 121.1 → m/z 74.1). Negative mode: CSA (m/z 152 → m/z 88; 17), [$^{13}$C$_3$, $^{15}$N]-CSA (m/z 156 → m/z 92; 17), GS-SO$_3^-$ (m/z 386 → m/z 306; 20). All peaks were manually integrated using Thermo Xcaliber Qual Browser. The quantification of metabolites was calculated by an isotope ratio-based approach according to published methods (*Bennett et al., 2008*).

## GC-MS-based selective ion monitoring (SIM)

For the [$^{13}$C$_5$]-glutamine → citrate labeling analysis, the TMS-derivatized samples were analyzed using previously established GC-MS conditions (*Kind et al., 2009*). The inlet temperature was set to 250°C and the inlet pressure was 11.4 psi. The purge flow was 3 mL/min and the split ratio was 2:1. The stationary phase was an Agilent J&W DB-5MS +10m Duraguard Capillary Column (30 m × 250 μm × 0.25 μm) (Part number: 122-5532G, Agilent Technology, Santa Clara, CA). The oven temperature gradient condition was set as following: 0 to 27.5 min, linear gradient from 60 to 325°C; 27.5 to 37.5 min, 325°C. The EI-MS scan range was 50 to 600 m/z and the scan speed was 2.7 scans/sec. The solvent delay was set to 5.90 min. The MSD transfer line temperature was 250°C, The EI energy was 70 eV, the EI source temperature was 230°C, and the MS Quadrupole temperature was 150°C. The ions for Selective Ion Monitoring (SIM) of derivatized citrate (4TMS) were as following: 465 [M]$^+$, 466 [M+1]$^+$, 467 [M+2]$^+$, 468 [M+3]$^+$, 469 [M+4]$^+$, 470 [M+5]$^+$, and 471 [M+6]$^+$. All peaks were manually integrated using Agilent MassHunter Qualitative Analysis Software and the natural abundance isotopes correction was performed according to published methods (*Mullen et al., 2011*) using IsoCor software (*Millard et al., 2012*).

## Cell viability assays

Cells were plated in 96-well plates at a density of 5,000–10,000 cells/well in 100 μL final volume. The next day, the medium was changed to 100 μL medium containing CSA, Na$_2$SO$_3$ or HTAU at the indicated concentrations. 3–5 days later, CellTiter-Glo (Promega) was used to measure cell viability. Alternatively, cells were fixed with 4% paraformaldehyde, stained with crystal violet, washed and dried. Crystal violet was solubilized in 10% acetic acid and the OD$_{600}$ was measured. Relative cell number was normalized to untreated cells. Experiments were repeated more than twice.

## Cell proliferation assays

Prior to the start of the proliferation assay, cells were treated with 0.25 μg/mL doxycycline for 2 days to induce CDO1 or GFP expression. Cells were seeded at 5,000 cells/well in 96-well plates and

allowed to proliferate for the indicated time points. Plates were fixed with 4% paraformaldehyde, stained with crystal violet, washed and dried. Crystal violet was solubilized in 10% acetic acid and the $OD_{600}$ was measured.

## Immunoblotting

Lysates were prepared in RIPA buffer (20 mM Tris-HCl [pH 7.5], 150 mM NaCl, 1 mM EDTA, 1 mM EGTA, 1% NP-40, 1% sodium deoxycholate) containing protease inhibitors (Roche complete). Protein concentrations were determined by the DC protein assay (Bio-Rad). Lysates were mixed with 6X sample buffer containing β-ME and separated by SDS-PAGE using NuPAGE 4–12% Bis-Tris gels (Invitrogen), followed by transfer to 0.45 μm Nitrocellulose membranes (GE Healthcare). The membranes were blocked in 5% non-fat milk in TBST, followed by immunoblotting. During this study, the Abcam CDO1 antibody was discontinued. The CDO1 antibody from Sigma Aldrich provides similar results for human and mouse CDO1 (data not shown).

## NADPH/NADP$^+$ assay

NSCLC cells were plated in 6 cm dishes and analyzed at 70% confluence. 4 hr prior to harvest, cell culture medium was changed to fresh media. Cells were extracted and the NADPH/NADP$^+$ ratio was measured using the NADP/NADPH-Glo Assay kit (Promega) according to the manufacturer's instructions.

## Protein synthesis assay

Protein synthesis analysis was performed using the Click-It HPG Alexa Fluor 488 Protein Synthesis Assay Kit (Life Technologies). Briefly, cells were treated with doxycycline (0.25 μg/mL) for 24 hr, then seeded into black walled, clear bottom 96-well tissue culture plates at 20,000 cells/well in doxycycline-containing media. The following day, cells were treated with HPG (1 μg/mL) in methionine-free DMEM (10% dFBS, 1% Pen/Strep, 10 mM HEPES). Cycloheximide (50 μg/mL) treatment was included as a positive control. After 4 hr, the cells were fixed with 3.7% formaldehyde in PBS for 15 min at room temperature, and stained according to the manufacturer's instructions. The fluorescence signal (ex/em: 475/520 nm) was measured using a Glomax plate reader (Promega), followed by staining with crystal violet. The fluorescence signal was normalized to the crystal violet absorbance value.

## Soft agar assays

Soft agar assays were performed in triplicate in 6-well dishes. A 1 mL base layer of 0.8% agar in RPMI was plated and allowed to solidify, then 5,000 cells/well were plated in 0.4% agar on top. The following day, 1 mL of RPMI was added to each well, and changed as needed. Colonies were allowed to form for 10–14 days, and wells were stained with 0.01% crystal violet in a 4%PFA/PBS solution. Plates were scanned on a flatbed scanner and colonies quantified with Image J.

## Statistical analysis

Data were analyzed using a two-sided unpaired Student's t test unless otherwise noted. GraphPad Prism seven software was used for all statistical analyses, and values of $p<0.05$ were considered statistically significant (*$p<0.05$; **$p<0.01$; ***$p<0.001$, ****$p<0.0001$. In some figures with multiple comparisons # is used in addition to *). The mean ± standard error of experiments performed in at least triplicate is reported. Similar variances between groups were observed for all experiments. Normal distribution of samples was not determined.

## Acknowledgements

We thank Florian Karreth and Isaac Harris for critical reading of the manuscript and Sae Bom Lee for laboratory assistance. GMD is supported by grants from the NIH (R37-CA230042), American Lung Association (LCDA-498544), the Miles for Moffitt Milestone Award, and the American Cancer Society's Institutional Research Grant. CCD is supported by NIH grant R00-CA194314. The Proteomics/Metabolomics Core is supported in part by the NCI (P30-CA076292), Moffitt Foundation, and a Florida Bankhead-Coley grant (06BS-02–9614).

## Additional information

### Funding

| Funder | Grant reference number | Author |
| --- | --- | --- |
| National Cancer Institute | R37-CA230042 | Gina DeNicola |
| American Lung Association | LCDA-498544 | Gina DeNicola |
| Moffitt Cancer Center | Milestone Award | Gina DeNicola |
| American Cancer Society | Institutional Research Grant | Gina DeNicola |
| National Cancer Institute | R00-CA194314 | Christian C Dibble |

The funders had no role in study design, data collection and interpretation, or the decision to submit the work for publication.

### Author contributions

Yun Pyo Kang, Conceptualization, Investigation, Methodology, Writing—original draft, Writing—review and editing; Laura Torrente, Investigation, Writing—review and editing; Aimee Falzone, Cody M Elkins, Investigation; Min Liu, Resources, Writing—review and editing; John M Asara, Methodology; Christian C Dibble, Funding acquisition, Investigation, Methodology, Writing—review and editing; Gina M DeNicola, Conceptualization, Supervision, Funding acquisition, Methodology, Writing—original draft, Writing—review and editing

### Author ORCIDs

Gina M DeNicola ⬤ https://orcid.org/0000-0001-6611-6696

### Ethics

Animal experimentation: Mice were housed and bred in accordance with the ethical regulations and approval of the IACUC (protocol # R IS00003893). Lung tumor formation was induced by intranasal installation of 2.5 x 107 PFU adenoviral-Cre (University of Iowa) as described previously (Jackson et al., 2001). Viral infections were performed under isofluorane anesthesia, and every effort was made to minimize suffering.

### Decision letter and Author response

Decision letter https://doi.org/10.7554/eLife.45572.042
Author response https://doi.org/10.7554/eLife.45572.043

## Additional files

### Supplementary files

• Transparent reporting form
DOI: https://doi.org/10.7554/eLife.45572.040

### Data availability

All data generated or analysed during this study are included in the manuscript and supporting files. Source data files have been provided for all figures.

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
