## [Decision Letter]

Thank you for submitting your article "Cysteine dioxygenase 1 is a metabolic liability for non-small cell lung cancer" for consideration by *eLife*. Your article has been reviewed by Sean Morrison as the Senior Editor, a Reviewing Editor, and three reviewers. The following individuals involved in review of your submission have agreed to reveal their identity: Mara Sherman (Reviewer #3).

The reviewers have discussed the reviews with one another and the Reviewing Editor has drafted this decision to help you prepare a revised submission.

Summary:

This study examines the effects of NRF2 activation on cell metabolism, focusing on its impact on cysteine metabolism. It finds that NRF2 promotes cystine uptake, leading to cysteine increases in cells and engagement of cysteine breakdown pathways involving CDO1. This has the consequence of producing the toxic products CSA and sulfite. As a result, they argue that loss of CDO1 expression in NRF2 active cancer cells, primarily through promoter methylation, occurs to suppress the formation of these toxic byproducts. Loss of CDO1 appears to be strongly selected for in cancers with KEAP1 mutations (which activate NRF2) and provides a mechanism to explain a metabolic adaptation and may have therapeutic implications.

The reviewers were generally positive about publication of this study, but felt consideration of the following points would improve the manuscript. Most involve text changes, but there are a couple points that should be experimentally addressed, if possible.

Essential revisions:

1) The authors propose that CDO1 promoter methylation and suppression in gene expression is more prevalent in KEAP1 mutant tumors. The data in Figure 3A,B are not very convincing since it appears that all lung tumors (KEAP1 WT or MUT) have decreased CDO1 expression. At a minimum, please discuss how CDO1 loss could benefit tumors that do not have KEAP1 mutations. Some additional work may also shed light on this question. For example, is it possible that many of the WT tumors have NRF2 activation through another mechanism? The authors could use existing NRF2 gene expression signatures (e.g. Goldstein, 2016) to assess the correlation of CDO1 loss with the enrichment of NRF2-driven transcriptional signatures in lung cancer. One reviewer also questioned whether NRF2 stabilization by ROS-dependent post-translational stabilization of NRF2 in KEAP1 WT lung cancer cells might mediate the same level of toxicity of CDO1 re-expression as seen in KEAP1 MUT lung cancer cells? This would be very interesting because it would suggest that the proposed mechanism described in this paper could occur in cancer types with exposure to high levels of ROS and therefore NRF2 activation.

2) Evidence of a putative growth-suppressive role for CDO1 in lung cancer is not as strong. Cystine tracing results are somewhat subtle (Figure 2E,F,H,I) and suggest that CDO1 is only partly responsible for directing CYS fate into the taurine synthesis pathway and for regulating GSH production in KEAP1-mutant cells, as part of a broader metabolic program. This is further reflected in the GSH, HTAU, and TAU measurements in CDO1 WT vs mutant lung cancer cells in Figure 4C, and the modest change in GSH levels in Figure 5A compared to other relevant metabolites. Importantly, though, CDO1 seems to play an important role in CSA production, and the authors demonstrate that CSA can phenocopy WT CDO1, suppressing cell proliferation and increasing sensitivity to lipid peroxidation. Together, it seems that the data in the study strongly link CDO1 to (CYS)2 uptake and to production of CSA, but further links to the taurine synthesis pathway and impacts on GSH levels are less clear. Revising of language throughout the manuscript to reflect possible alternative mechanisms and to leave room for roles of additional, perhaps compensatory pathways in regulation of cellular GSH levels in this context would be helpful. This is mentioned in the Discussion section in relation to cancer cells, but as effects of CDO1 on GSH production in MEFs are quite subtle, the proposed connection between CDO1 and GSH should be revised throughout the text.

3) There was a concern raised that complete depletion of cystine (Figure 5—figure supplement 4B) is not physiological and will lead to robust growth suppression in most cells. To further test the proposed mechanism by which the increased uptake of cystine through XCT stabilized CDO1 leading the toxic byproducts, the authors should assess whether pharmacologic or genetic loss of XCT would suppress toxicity by CDO1 complementation. Furthermore, they can also modulate cystine levels in the media to regulate the uptake of cystine by XCT and thus impact intracellular cystine levels and sensitivity to CDO1 re-expression.

4) Based on many prior studies, KEAP1 mutant cells produce high levels of glutathione and NADPH which create a highly reduced intracellular environment. Furthermore, based on recent data (Cao et al., 2019) NRF2 induces transporters that enable the secretion of glutathione. It is therefore possible that the reactive cysteines of glutathione would also react with the CDO1/GOT1 generated sulfite molecules both intracellularly and extracellularly. Measurement of GSH/GSSG ratios upon re-expression of CDO1 are likely present within their existing LC-MS data. If so, can this be reported? A decrease in the ratio would also be in line with the fact that NADPH/NADP ratio decreases in response to CDO1 re-expression.

5) Regarding the ideas in Figure 7 that the supply of cysteine via import of cystine and its reduction by NADPH-dependent enzymes like thioredoxin reductase or glutathione reductase (the thioredoxin or glutaredoxin systems) the authors should discuss the possibility of the transsulfuration pathway from methionine. An article showing the importance of this pathway can be found in: Prigge et al., 2017.

6) One reviewer was concerned about the lack of any data to support the importance of this mechanism in vivo. They felt transplants using some of the human lung cancer cell lines to assess whether re-expression of WT CDO1 suppresses tumor growth as opposed to re-expression of mutant inactive CDO1 would help the paper. I agree this is a nice experiment, but is not essential for publication in *eLife*. If the data could be provided in a reasonable time frame it would improve the paper. If providing these data is not possible, then please discuss that the lack of in vivo data is a limitation of the study and mention that this work will be needed to further test the model in the future.

---

## [Author Response]

Essential revisions:1) The authors propose that CDO1 promoter methylation and suppression in gene expression is more prevalent in KEAP1 mutant tumors. The data in Figure 3A,B are not very convincing since it appears that all lung tumors (KEAP1 WT or MUT) have decreased CDO1 expression. At a minimum, please discuss how CDO1 loss could benefit tumors that do not have KEAP1 mutations. Some additional work may also shed light on this question. For example, is it possible that many of the WT tumors have NRF2 activation through another mechanism? The authors could use existing NRF2 gene expression signatures (e.g. Goldstein, 2016) to assess the correlation of CDO1 loss with the enrichment of NRF2-driven transcriptional signatures in lung cancer.

We have used our own NRF2 gene expression signature (DeNicola et al., 2015) to examine the association of CDO1 methylation with NRF2 activation. We find that KEAP1 mutation-independent activation of NRF2 accounts for an additional fraction of the CDO1 methylated patient population, and when combined with the KEAP1 mutant population these “NRF2 high” patients are still significantly enriched for CDO1 methylation (new Figure 3—figure supplement 1C). However, a significant fraction of the CDO1 methylated patients remain unaccounted for by NRF2 activation. This finding is consistent with the findings from our cell line studies, in which a fraction of the KEAP1 wild-type lines were found to accumulate significant cysteine in a NRF2- and xCT-independent manner (Figure 3D). These lines accumulated CDO1 and were sensitive to its anti-proliferative effects. Thus, our findings support that NRF2 activation is only one mechanism by which cells can accumulate intracellular cysteine, providing an explanation for how CDO1 loss could benefit KEAP1 wild-type (and NRF2 low) tumors that have high intracellular cysteine.

One reviewer also questioned whether NRF2 stabilization by ROS-dependent post-translational stabilization of NRF2 in KEAP1 WT lung cancer cells might mediate the same level of toxicity of CDO1 re-expression as seen in KEAP1 MUT lung cancer cells? This would be very interesting because it would suggest that the proposed mechanism described in this paper could occur in cancer types with exposure to high levels of ROS and therefore NRF2 activation.

This reviewer raises a very interesting point. To address this point, we used the superoxide-generating compound β-lapachone, which induces NRF2 stabilization in the KEAP1 wild-type cell lines H1299 and H1975 (new Figure 3—figure supplement 2A). We find that NRF2 stabilization by β-lapachone promotes the accumulation of intracellular cysteine, CDO1 stabilization, the production of CDO1-dependent metabolites CSA and Cys-SO3, and cystine depletion (new Figure 3—figure supplement 2B, Figure 5H-J). Thus, chronic oxidative stress in the tumor microenvironment may also select for CDO1 loss.

2) Evidence of a putative growth-suppressive role for CDO1 in lung cancer is not as strong. Cystine tracing results are somewhat subtle (Figure 2E,F,H,I) and suggest that CDO1 is only partly responsible for directing CYS fate into the taurine synthesis pathway and for regulating GSH production in KEAP1-mutant cells, as part of a broader metabolic program. This is further reflected in the GSH, HTAU, and TAU measurements in CDO1 WT vs mutant lung cancer cells in Figure 4C, and the modest change in GSH levels in Figure 5A compared to other relevant metabolites. Importantly, though, CDO1 seems to play an important role in CSA production, and the authors demonstrate that CSA can phenocopy WT CDO1, suppressing cell proliferation and increasing sensitivity to lipid peroxidation. Together, it seems that the data in the study strongly link CDO1 to (CYS)2 uptake and to production of CSA, but further links to the taurine synthesis pathway and impacts on GSH levels are less clear. Revising of language throughout the manuscript to reflect possible alternative mechanisms and to leave room for roles of additional, perhaps compensatory pathways in regulation of cellular GSH levels in this context would be helpful. This is mentioned in the Discussion section in relation to cancer cells, but as effects of CDO1 on GSH production in MEFs are quite subtle, the proposed connection between CDO1 and GSH should be revised throughout the text.

We agree with the reviewers and have added/modified these points in the text. First, only modest effects on GSH are observed in MEFs (modestly is added to subsection “Nrf2 promotes the entry of cysteine into the taurine synthesis pathway via Cdo1 in non-transformed, primary MEFs”and the Discussion section), which may be due to incomplete CDO1 deletion by CRISPR/Cas9, feedback inhibition of GCLC by GSH, or other mechanisms. Further, the reviewer is correct that CDO1 is only partly responsible for directing CYS into taurine, as the alternative route of CYS entry into the taurine synthesis pathway via CoA breakdown is CDO1-independent and cannot be distinguished from CDO1-dependent taurine synthesis (subsection “Nrf2 promotes the entry of cysteine into the taurine synthesis pathway via Cdo1 in non-transformed, primary MEFs”). Second, we agree that CDO1 plays an important role in cystine uptake and CSA production, but has a minor role in taurine synthesis and GSH production in NSCLC cells (Discussion section). These points have been added to the text.

3) There was a concern raised that complete depletion of cystine (Figure 5—figure supplement 4B) is not physiological and will lead to robust growth suppression in most cells. To further test the proposed mechanism by which the increased uptake of cystine through XCT stabilized CDO1 leading the toxic byproducts, the authors should assess whether pharmacologic or genetic loss of XCT would suppress toxicity by CDO1 complementation. Furthermore, they can also modulate cystine levels in the media to regulate the uptake of cystine by XCT and thus impact intracellular cystine levels and sensitivity to CDO1 re-expression.

We agree that complete depletion of cystine is unlikely to occur under physiological conditions where cells are adequately perfused. Importantly, we find that sulfite production and cystine starvation are not necessary for the growth suppressive functions of CDO1 (Figure 6 and Figure 7, GOT1 KO cells). We perform most experiments in our manuscript four hours after media replenishment to avoid the secondary effects of starvation raised by the reviewer. The objective of the experiment in Figure 5—figure supplement 4B) is to interrogate whether cystine starvation is uniformly toxic to NSCLC cells, as the reviewer points out, regardless of KEAP1 status. We have clarified this in the text (subsection “CDO1 restoration in NSCLC cells promotes sulfite production, thereby depleting cystine via sulfitolysis”). The data in Figure 5—figure supplement 4B support that the toxicity of CDO1 is not a consequence of some cell lines having increased tolerance to cystine starvation or sulfite than others.

We thank the reviewers for the suggestion to use xCT inhibition and cystine modulation to further support our findings regarding CDO1 toxicity. To this end, we have first used a sublethal concentration of erastin (0.5uM), which was previously demonstrated to rescue the effects of glutaminase inhibition in KEAP1 mutant cells (Sayin et al., 2017), and have confirmed that cystine uptake is inhibited under these conditions (Figure 5—figure supplement 1E). In agreement with the post-translational stabilization of CDO1 by cysteine, we find that erastin treatment leads to complete loss of CDO1 expression in H460 and A549 cells and completely rescues the CDO1 proliferation defect (Figure 5—figure supplement 1D,F). We further cultured the cells in 10uM cystine, which produced similar results (Figure 5—figure supplement 1D,F). Collectively, these results support role of cysteine in the stabilization of CDO1 and its growth suppressive effects.

4) Based on many prior studies, KEAP1 mutant cells produce high levels of glutathione and NADPH which create a highly reduced intracellular environment. Furthermore, based on recent data (Cao et al., 2019) NRF2 induces transporters that enable the secretion of glutathione. It is therefore possible that the reactive cysteines of glutathione would also react with the CDO1/GOT1 generated sulfite molecules both intracellularly and extracellularly. Measurement of GSH/GSSG ratios upon re-expression of CDO1 are likely present within their existing LC-MS data. If so, can this be reported? A decrease in the ratio would also be in line with the fact that NADPH/NADP ratio decreases in response to CDO1 re-expression.

In agreement with prior studies (Menzel et al., 1986), we do find that sulfite is capable of reacting with GSSG with similar kinetics to cystine, when an excess of sulfite is mixed with GSSG in the absence of cells (New Figure 5—figure supplement 2C,D). However, as the reviewer states, we observe that KEAP1 mutant cells have a highly reduced intracellular environment. While intracellular GSH concentrations are in the 10-20mM range, intracellular GSSG levels are approximately 20uM and undetectable extracellularly (New Figure 5—figure supplement 2E,F). It is important to note that sulfite would react with GSSG, not GSH, and there is little GSSG available to react with sulfite, in contrast to cystine, which is approximately 200um intracellularly and from 200um-50um extracellularly, depending on how long the cells are cultured (New Figure 5—figure supplement 2E,F). Importantly, through our quantitative analysis of cystine and downstream metabolites, most CDO1-dependent cystine is accounted for (Figure 6B), suggesting that if sulfite does react with glutathione it is not a major fate of cysteine in our system. However, the levels of GSSG in the extracellular tumor microenvironment may exceed that of cystine (Sullivan et al., 2019), and we have added this point to the Discussion section.

Regarding the relationship between the NADPH/NADP^+^ ratio and the GSH/GSSG ratio, we examined our existing quantitative GSH and GSSG data. While we did not have these data from the GOT1 KO cells, we did have it from our KEAP1 complementation experiment for A549, H460 and H1944. We find that under non-stressed conditions CDO1 expression does not dramatically affect GSH or GSSG (new Figure 7B). The concomitant decrease in NADPH-dependent processes including proline synthesis and reductive carboxylation suggests that NADPH is more limiting for these processes than glutathione reduction. This is consistent with a low Km of glutathione reductase for NADPH (8uM, Brenda) and a high Km of PYCR1 (proline synthesis) for NADPH (300uM, Brenda). This is also consistent with recent work from Zou et al., which finds that the NADPH/NADP^+^ ratio fluctuates throughout the cell cycle to support proliferation without any change in ROS. However, under oxidizing conditions, Zou et al. find that NADPH becomes significantly depleted and thiols become oxidized. Therefore, NADPH may only be limiting for GSSG reduction following oxidant exposure.

5) Regarding the ideas in Figure 7 that the supply of cysteine via import of cystine and its reduction by NADPH-dependent enzymes like thioredoxin reductase or glutathione reductase (the thioredoxin or glutaredoxin systems) the authors should discuss the possibility of the transsulfuration pathway from methionine. An article showing the importance of this pathway can be found in: Prigge et al., 2017.

We have previously found that the transsulfuration pathway contributes very little to the cysteine pool under normal conditions in KEAP1 mutant cells (DeNicola et al., 2015), where we examined labeling in the CYS residue of glutathione from serine, and we similarly observe complete labeling of intracellular cysteine from 13C-cystine in A549 and H460 cells (not shown). However, it is possible that the transsulfuration pathway may significantly contribute to the cysteine pool in the NRF2- and xCT- low / cysteine high cell lines in Figure 3D, where the high intracellular cysteine levels are not currently explained. This point has been added to the Discussion section.

*6) One reviewer was concerned about the lack of any data to support the importance of this mechanism* in vivo*. They felt transplants using some of the human lung cancer cell lines to assess whether re-expression of WT CDO1 suppresses tumor growth as opposed to re-expression of mutant inactive CDO1 would help the paper. I agree this is a nice experiment, but is NOT essential for publication in eLife. If the data could be provided in a reasonable time frame it would improve the paper. If providing these data is not possible, then please discuss that the lack of* in vivo *data is a limitation of the study and mention that this work will be needed to further test the model in the future.*

We agree that in vivo data would strengthen the paper, and we have focused our in vivo experiments on GEMM models given the generally poor perfusion and inadequate vasculature of xenograft models, which may influence cystine levels in the tumor microenvironment. To this end, we have generated K-Ras^G12D^; p53^flox/flox^ lung tumor mice expressing either wild-type KEAP1, heterozygous for KEAP1^R554Q^, or homozygous for KEAP1^R554Q/R554Q^. We chose this model because KEAP1 deletion in the K-Ras^G12D^; p53^flox/flox^ lung tumor model was recently shown to increase NRF2 activity and promote cystine uptake (Romero et al., 2017). We find that KEAP1 loss of function leads to a gradient of NRF2 activation with KEAP1^W/WT^ tumors expressing little to no NQO1. KEAP1^R554QWT^ tumors displayed increased NQO1 expression, and KEAP1^R554Q/R554Q^ tumors were strongly positive. KEAP1^R554Q/WT^ tumors were modestly larger than KEAP1^WT/WT^ tumors, but surprisingly KEAP1^R554Q/R554Q^ tumors were very small and low grade. Importantly, KEAP1^R554Q/R554Q^ tumors strongly expressed CDO1, demonstrating that NRF2 activation promotes CDO1 accumulation under physiological conditions in vivo, and suggesting that CDO1 may impede tumor progression. These new data are found in Figure 8.

To directly interrogate whether CDO1 impedes tumor progression, we have crossed a lox-STOP-lox-Cas9 allele (Chiou et al., 2015) into our model to delete CDO1 by CRISPR. However, these experiments are still ongoing and therefore, we have added a section to both the results and the discussion regarding the additional in vivo work needed to determine whether CDO1 mediates the block in tumorigenesis in the KEAP1^R554Q/R554Q^ model (Discussion section).